# Single cell transcriptomic profiling of a neuron-astrocyte assembloid tauopathy model

Hannah Drew Rickner [1,9], Lulu Jiang [2,9], Rui Hong [3], Nicholas K. O'Neill[3], Chromewell A. Mojica [4], Benjamin J. Snyder[4], Lushuang Zhang [2], Dipan Shaw [5], Maria Medalla[4,6], Benjamin Wolozin [2,6,7] ✉ & Christine S. Cheng [1,3,5,8] ✉

The use of iPSC derived brain organoid models to study neurodegenerative disease has been hampered by a lack of systems that accurately and expeditiously recapitulate pathogenesis in the context of neuron-glial interactions. Here we report development of a system, termed AstTau, which propagates toxic human tau oligomers in iPSC derived neuron-astrocyte assembloids. The AstTau system develops much of the neuronal and astrocytic pathology observed in tauopathies including misfolded, phosphorylated, oligomeric, and fibrillar tau, strong neurodegeneration, and reactive astrogliosis. Single cell transcriptomic profiling combined with immunochemistry characterizes a model system that can more closely recapitulate late-stage changes in adult neurodegeneration. The transcriptomic studies demonstrate striking changes in neuroinflammatory and heat shock protein (HSP) chaperone systems in the disease process. Treatment with the HSP90 inhibitor PU-H71 is used to address the putative dysfunctional HSP chaperone system and produces a strong reduction of pathology and neurodegeneration, highlighting the potential of AstTau as a rapid and reproducible tool for drug discovery.

Tauopathies are characterized by the accumulation of misfolded and hyperphosphorylated tau which form neurofibrillary tangles and precipitate neurodegeneration. These intracellular factors such as aggregation of tau and impaired proteostasis contribute to the tauopathy degenerative process, typically in combination with inflammation and vascular dysfunction[1–7]. The resulting neuronal damage and tau pathology is closely associated with cognitive decline in tauopathies[8,9]. This suggests that broadening our understanding of tau pathogenesis will enable the development of effective therapeutics against tauopathies.

Many models for tauopathies have been created, but few if any accurately reflect the human milieu. Animal models develop tangles but lack a human background, often producing results that do not translate successfully to human disease-modifying strategies[10,11]. Increasingly, investigators are turning to human induced pluripotent stem cells (hiPSCs) to overcome such challenges, but such 2D models typically exhibit limited arborization, neuron–neuron, neuron–glial interactions, and the extensive tau pathology and degeneration present in tauopathies[12–25]. Three-dimensional (3D) organoid models of tauopathies show great promise, but most require maturation over an

[1]Department of Biology, Boston University, Boston, MA 02215, USA. [2]Department of Pharmacology and Experimental Therapeutics, Boston University School of Medicine, Boston, MA 02118, USA. [3]Program in Bioinformatics, Boston University, Boston, MA 02215, USA. [4]Department of Anatomy & Neurobiology, Boston University School of Medicine, Boston, MA 02118, USA. [5]Informatics Group, J. Craig Venter Institute, La Jolla, CA 92037, USA. [6]Department of Neurology, Boston University School of Medicine, Boston, MA 02118, USA. [7]Center for Systems Neuroscience, Boston University, Boston, MA 02118, USA. [8]Department of Psychiatry, University of California San Diego, La Jolla, CA 92093, USA. [9]These authors contributed equally: Hannah Drew Rickner, Lulu Jiang. ✉e-mail: bwolozin@bu.edu; csc003@ucsd.edu

extended time frame of months or years, lack an explicit ratio of neuron and glial compartments, and develop limited pathology and neurodegeneration[26–35]. This difficulty in producing an expedient human organoid model combined with the inadequacies of current animal models for tauopathies has stymied the development of effective research and therapeutics.

To address this challenge, we present a model in which hiPSC-derived neurons are seeded with human tau oligomers, combined with hiPSC-derived astrocytes, and grown in spheroidal culture to create a 3D assembloid model termed "asteroids"[36,37]. The resulting neurons and astrocyte disease model rapidly recapitulates physiologically relevant spatial cellular interactions and exhibits advanced maturation over other cerebral organoid or spheroid models[36–38]. Increasing evidence indicates that tau propagation is an important contributor to multiple human neurodegenerative diseases[39–42]. Accumulating studies demonstrate that exogenous tau oligomers can be internalized, trafficked, and propagated in animal and cell models, spreading disease pathology in a prion-like manner[43–53]. We show that challenging the asteroid model with toxic human oligomeric tau (oTau) yields robust tau pathology, neurodegeneration and reactive astrogliosis over just 21 days of 3D culture (DIV3D). We will refer to this system as AstTau. A key to the success of this model lies in seeding with tau oligomers, and subsequent expansion of the pathology through propagation to other neurons. Most studies of tau propagation have focused on tau fibrils, which propagate pathology but elicit little if any neurodegeneration[43–52]. In contrast, we and others have shown that exposure to tau oligomers propagates tau pathology and induces robust pathology with advanced neurodegeneration[22,54–56]. The tauopathy asteroid system AstTau incorporates this propagation of toxic tau oligomers to create a highly reproducible human 3D model of tauopathy in just 3 weeks. The AstTau system enables analysis of neurodegenerative processes while also enabling concomitant analysis of the disease responses of astrocytes, which are increasingly noted to play an important yet understudied role in the development of tauopathies[37,57–62]. In addition, the single cell RNA sequencing (scRNA-seq) presented below provides detailed transcriptomic profiling of cell-specific responses to the pathological progression of tauopathy and enables hypothesis testing using the AstTau system.

## Results

### 3D asteroids rapidly recapitulate neuronal and astrocytic signatures

To establish a 3D assembloid system of neuron and astrocyte coculture for modeling neurodegenerative disease hiPSC-derived neural progenitor cells (NPCs) were directed into neuronal and astrocytic lineages (Fig. 1a). The neuronal lineage was created by NGN2 (*NEUROG2*) overexpression to produce hiPSC-derived neuronal cells (hiNCs), while the astrocytic lineage was generated via small molecule-directed differentiation to produce hiPSC-derived astrocytes (hiACs) (Fig. 1a and Supplementary Fig. 1a). The resulting hiNCs and hiACs were then combined into 3D spheroidal culture termed asteroids as described in prior publication[37] (~2000 cells each) and allowed to develop over the course of 3 weeks (Fig. 1a, b and Supplementary Fig. 1b). Within the asteroids hiNCs expressed robust differentiation by immunolabeling of canonical neuronal markers including MAP2, Tuj1 (*TUBB3*), and NeuN (*RBFOX3*), synaptic markers synaptophysin (*SYP*) and PSD95 (*DLG4*), and excitatory and inhibitory neuronal markers VGLUT1 and GAD67 respectively (Fig. 1b and Supplementary Fig. 1c–e). hiACs similarly expressed canonical astrocyte markers such as S100β (*S100B*), EAAT1 (*SLC1A3*), vimentin (*VIM*), and GFAP (Fig. 1b and Supplementary Fig. 1c–e). Pre-differentiation of hiACs and hiNCs allowed for the development of this mature culture system that can rapidly reproduce physiological spatial interactions when neurons and astrocytes are combined in a 3D culture system (Supplementary Fig. 2a–c). 2D mixed cultures containing neuron and astrocytes do not show the

extensive arborization and intimate connections observed with 3D culture (Supplementary Fig. 2a–c). Incorporation of astrocytes was necessary for asteroid survival, with 3D cultures of neurons alone dying within 7 days (Supplementary Fig. 2b). Both neurons and astrocytes exhibited processes that were striking for their length and complexity, as well as for the intimate association (Supplementary Fig. 1b, c).

Asteroid cellular phenotypes were characterized by acquiring scRNA-seq profiles for 26,789 single cells with an average of 2000 genes per cell across a time course of 3 weeks (Fig. 1c–f and Supplementary Fig. 3). Visualization by uniform manifold approximation and projection (UMAP) demonstrated the separation of cells into trajectory guided clusters (Fig. 1c). These clusters were identified as the cycling neuronal progenitors (NEU_P), astrocytes precursors (ASC_P), astrocytes (ASC), inhibitory neurons (IN_NEU) and excitatory neurons (EX_NEU) (Fig. 1c) based on the expression of established cell-type-specific genes including *TOP2A* and *NES* for progenitors, *SLC1A3* (EAAT1), *VIM*, and *PEA15* for astrocytes, and *SNAP25*, *GAP43*, and *STMN2* for neurons (Fig. 1c, d and Supplementary Fig. 5). Neuronal clusters expressed synaptic markers transcripts including *DLG4* (PSD95) and *SYP* and were further separated by inhibitory neuronal markers including *GAD1* and *SLC32A1* and excitatory neuronal markers *SLC38A1* and *GRIA4* (Fig. 1c, d, i and Supplementary Fig. 5). Manual cell typing was further supported by the automated cell typing platform SingleR (Supplementary Fig. 5e). Notably, over 21 days of 3D coculture (21 DIV3D) populations of progenitors decreased while the populations of mature cell types showed a concomitant increase (Fig. 1e–h). Quantification of progenitor markers *SOX2* and *VIM* and mature neuronal marker *MAP2* decreased and increased respectively over the 21 DIV3D time course by both immunolabeling and scRNA-seq gene expression (Fig. 1g, h). These transcriptomic analyses further confirmed the robust differentiation previously observed by protein-based immunolabeling (Fig. 1b and Supplementary Fig. 1c–e). Cell-type composition across 5 replicate batches of asteroid cultures displayed notable consistency by immunolabeling and scRNA-seq cluster composition analysis (Supplementary Fig. 6). To compare the maturity of neurons and astrocytes in our asteroid culture system to traditional cerebral organoid models, we integrated our asteroid scRNA-seq data with an existing published scRNA-seq dataset of forebrain cerebral organoids generated using three previously published protocols with different levels of directed differentiation[38]. Clustering, cell-type identification, and cluster composition analysis were performed on the now integrated dataset and revealed that the asteroid model (red) was overrepresented in more mature cell-type clusters compared to the cerebral organoid model (blue) (Fig. 1j and Supplementary Fig. 7).

Electrophysiological profiling with whole-cell patch-clamp recordings revealed that asteroid hiNCs were biophysically functional, with synaptic currents and active conductances (Supplementary Fig. 4e, f). The asteroid neurons exhibited depolarized resting potentials ($V_{rest} = -40$ to $-50$ mV) and no recordable action potential (AP) spiking activity (Supplementary Fig. 4f–j), consistent with being somewhat electrophysiologically immature. However, recorded asteroid neurons had a prominent depolarizing sag potential ($V_{sag}$) in response to 2 s hyperpolarizing current injections (−100 pA) and showed a depolarizing hump or spikelet in response to large amplitude 2 s current injections (greater than +200 pA; Supplementary Fig. 4f). Further, these asteroid neurons had excitatory postsynaptic currents (EPSCs, Supplementary Fig. 4e) detected at low frequencies (<0.1 Hz) and were morphologically confirmed to have dendritic spines and axonal boutons/terminations (Supplementary Fig. 4b–d). These biophysical properties were consistent with channel expression profiles revealed via snRNA-seq. While hiNCs exhibited low expression of key voltage-gated sodium and potassium channels necessary for AP generation[63–65], they had high expression of hyperpolarization cyclic-nucleotide gated (HCN) channel subunits that mediate the Ih current

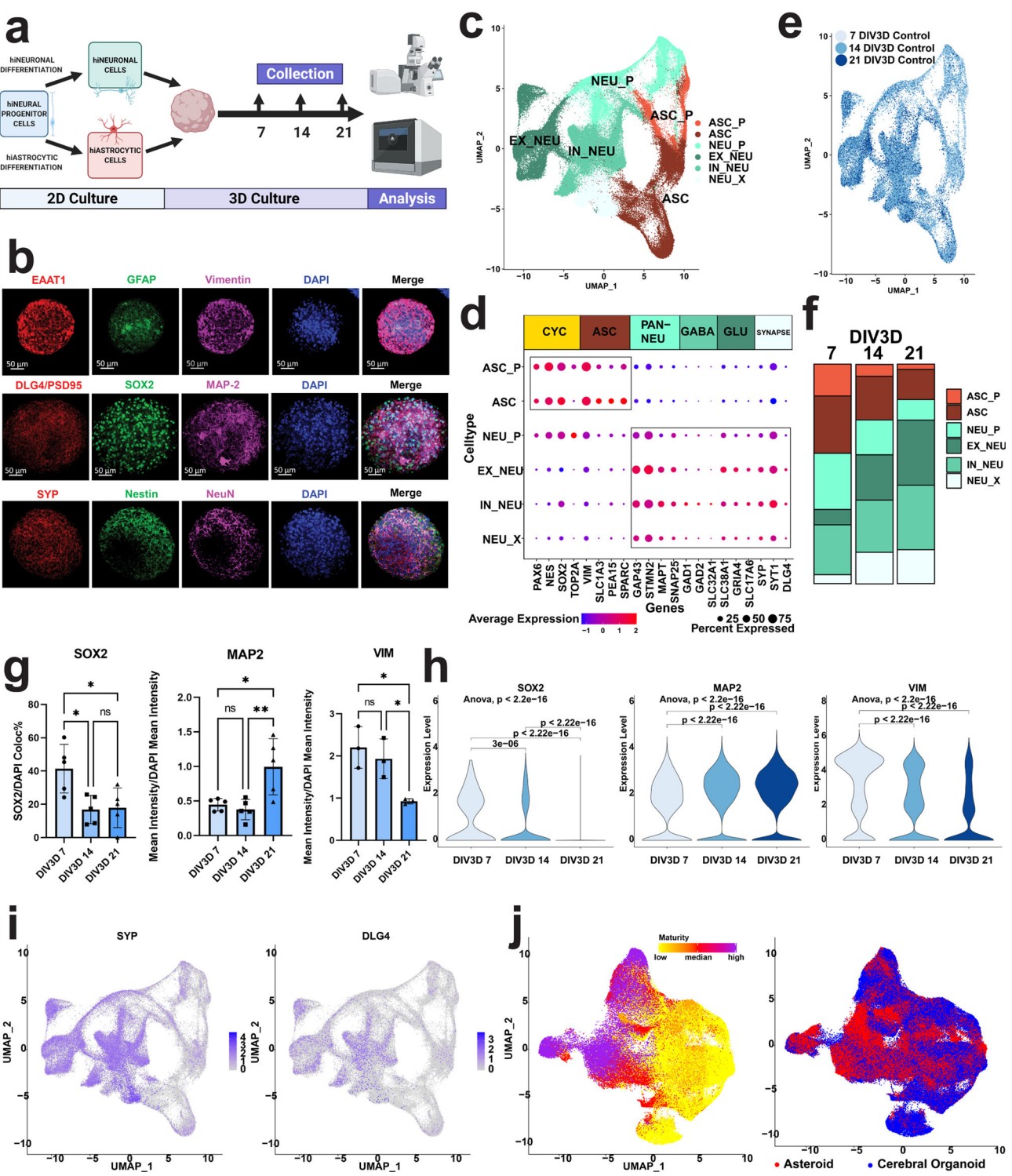

likely underlying the prominent depolarizing sag potential that we have observed (Supplementary Fig. 1f, g). Consistent with the presence of EPSCs, hiNCs also showed substantial expression of voltage-gated Ca2+ channels that are critical for neurotransmitter synaptic release (Supplementary Fig. 4h).

### oTau seeding induces rapid tau misfolding, aggregation, and neurodegeneration in the 3D neuron and astrocyte coculture AstTau model

We proceeded to model tauopathy in asteroid cultures (AstTau) (Fig. 2a). Human oligomeric tau (oTau) was isolated from 9-month-old PS19 P301S tau mice by centrifugation using a well-validated method[55,66,67]. The PS19 P301S tau mice model overexpresses human 4R1N P301S Tau and post-extraction, these human tau fractions were identified by native gel western blot with human-specific tau antibody Tau13 (Supplementary Fig. 8a).

The hiNCs were exposed to human oTau (0.04 mg/mL) for 24 h after which they were washed and combined with hiACs to generate the self-aggregating AstTau (Fig. 2a and Supplementary Fig. 8b). Analysis of the AstTau cultures over 21 DIV3D showed rapid development of tau pathology, including tau hyperphosphorylation, misfolding, oligomerization, and fibrilization (Figs. 2 and 3). Statistically significant increases were observed for misfolded tau (Fig. 2b–e). Misfolding and oligomerization of tau were observed using the MC1 and TOMA2

**Fig. 1 | 3D iPSC-derived neuron and astrocyte asteroid coculture rapidly recapitulates neuronal and astrocytic interactions and signatures. a** Experimental schematic diagraming 2D culture of hiPSC-derived neurons and astrocytes, 3D asteroid coculture, and immunolabeling and single-cell transcriptomic analysis at indicated timepoints. Created with BioRender.com. **b** Representative immunolabeling of 21 DIV3D asteroids with cell-type markers, including astrocytic precursor markers (VIM), astrocytic markers (EAAT1, GFAP), neuronal precursor markers (SOX2, NES), neuronal markers (MAP2, NEUN), and synaptic markers (DLG4/PSD95, SYP). Scale bar = 50 μm. **c** UMAP of 26,789 single-cell transcriptomic profiles by scRNA-seq after filtration and cell-type identification, including all cells from 7, 14, and 21 DIV3D (see "Methods"). Five main cell types are identified: excitatory neurons (EX_NEU, dark green), inhibitory neurons (IN_NEU, light green), neuron precursors (NEU_P, teal), astrocytes (ASC, dark red), and astrocyte precursors (ASC_P, light red). **d** Dot plot of key cell-type marker transcript expression including cycling (CYC), astrocytic (ASC), pan-neuronal (PAN-NEU), inhibitory neuronal (GABA), excitatory neuronal (GLU), and synaptic markers (SYNAPSE) in cell-type clusters. **e** UMAP projection colored by timepoint across the 7–21 days in 3D culture (DIV3D) time course (blue gradient) highlighting the model trajectory. **f** Percent composition of cell-type clusters across the 7–21 DIV3D time course, indicating a decrease in precursor cell proportions and a correlated increase in mature NEU and ASC proportions. **g, h** Quantification of select immunolabeled cell-type markers (**g**) and scRNA-seq transcript expression (**h**) across the 7–21 DIV3D time course (blue gradient), indicating a decrease in precursor markers (*SOX2, VIM*) and an increase in mature neuronal marker (*MAP2*). Data obtained from 5 independent asteroids. (ANOVA, ns: $P > 0.05$, *$P <= 0.05$, **$P <= 0.01$). **i** UMAP feature plot indicating gene expression of neuronal synaptic markers *DLG4* (PSD95) and *SYP*. **j** UMAP integration of the asteroid scRNA-seq dataset (Asteroid, red) with a cerebral organoid scRNA-seq dataset (Cerebral Organoid, blue[38]) highlighting overrepresentation of the asteroid cells in more mature cell clusters (see "Methods" and Supplementary Fig. 7).

antibodies, respectively, with each showing similar patterns of evolution (Fig. 2b–e and Supplementary Fig. 9a–d). Immunolabeling of 10 μM AstTau cryosections demonstrated TOMA2 localized selectively to Tuj1+ neurons (Supplementary Fig. 9i, j). Fibrillar tau pathology also evolved, but at a slower rate as seen in clinical tauopathy progression[68]. Fibrillar tau reactive with the dye thioflavin S (ThioS) became evident only at 21 DIV3D (Fig. 2f, g). In addition, hyperphosphorylated tau exhibiting corresponding increases at positions S202/5 and S262 (Fig. 3a, b and Supplementary Fig. 9e–h). This increase was additionally demonstrated by AT8 immunoblotting of AstTau at 21 DIV3D (Fig. 3c).

The striking propagated neuropathology was also associated with neurodegeneration by 21 DIV3D. Prominent neuronal injury at this late timepoint was evident by Fluoro Jade B labeling (Fig. 3e, g), reduced immunolabeling of Tuj1 and MAP2 (Supplementary Figs. 9a–d and 10a, b) and increased LDH release (Supplementary Fig. 10c). In addition, we observed reciprocal changes in BAX, a cell death regulator. Immunoblotting demonstrated decreases in full-length inactive BAX and corresponding increases in activated cleaved BAX with induction of pathology in the AstTau model at 21 DIV3D, indicative of activated cell death pathways (Fig. 3h–j). Notably, neurodegeneration in AstTau is associated with a loss of excitatory neurons as evidenced by scRNA-seq cell-type composition analysis and 25% loss of VGLUT1 immunolabeling, as is observed in clinical tauopathy progression (Supplementary Fig. 10d–f). There is also a striking reduction in synaptic gene markers by scRNA-seq in AstTau (Supplementary Fig. 10i).

Astrocytes also showed striking pathophysiological responses reminiscent of that observed in the human brain[57,58]. The astrocytic marker S100β and reactive astrogliosis marker GFAP showed statistically significant increases in reactivity beginning at 14 DIV3D and becoming prominent at 21 DIV3D (Fig. 3a, d–f). Astrocytic processes in AstTau cultures showed greater area and more intimate localization with regions of neuronal damage than in control asteroid cultures (Fig. 3a). Taken together, these results indicate that over 21 DIV3D, the AstTau system develops a range of responses analogous to those observed in the brains of human subjects with tau-mediated neurodegeneration[5,6,57,58].

A requirement for oTau in the seeding process was demonstrated by immunodepleting oTau, which prevented any subsequent tau pathology or neurodegeneration (Supplementary Fig. 8d). The role of seeded oTau was further explored by exposing hiNCs to FITC labeled oTau. FITC labeled oTau was present in >65% of hiNCs 24 h after seeding, however, the vast majority of the seeded tau was degraded by 5 DIV3D (Supplementary Fig. 8b–d), and remaining FITC showing no correlation with accumulating tau pathology (Supplementary Fig. 8d). These results indicate that tau pathology developing after 5 DIV3D was generated largely from endogenous tau sources. In addition, propagation of oTau-seeded pathology was demonstrated by merging unlabeled (GFP−) assembloids that had been seeded with oTau (at the 2D stage) with GFP+ labeled assembloids that had not been exposed to oTau. In the combined mixed assembloids misfolded and oligomerized tau was found in GFP− and GFP+ neurons, indicating propagation of pathology from oTau-seeded neurons to unexposed neurons (Fig. 3k). In addition, quantification of the accumulation of immunolabeled MC1 showed the same level of misfolded tau pathology developed in pure oTau-seeded cultures and mixed cultures compared to control cultures (Fig. 3l, m). Colocalization of the NGN2-GFP and MC1 signal returned a Pearson's R coefficient of correlation of 0.09 as compared to a 0.89 correlation of Tuj1 and MC1 signal in the same assembloids, further validating that a subset of unexposed GFP+ neurons were subject to oTau propagation (Fig. 3n).

Further control studies demonstrated that the observed pathology in AstTau was selective for oTau by examining the development of pathology in assembloids exposed to preformed α-synuclein fibrils (PFF-αSYN) (Supplementary Figs. 11 and 12). While PFF-αSYN exposure (1 μg/mL) resulted in the accumulation of phosphorylation α-synuclein (Supplementary Fig. 12d, e), it did not elicit oTau-specific pathology, such as the accumulation of toxic tau as marked by TOMA2 and MC1 (Supplementary Fig. 12a, b, d, f), hyperphosphorylated tau as marked by p202/AT8 (Supplementary Fig. 12i, j), fibrillization or reactive astrogliosis (Supplementary Fig. 12a, c, d, g), or neurodegeneration (Supplementary Fig. 12i, k). Together these data support a direct role for the human toxic tau oligomers in the development of oTau-specific pathology in the AstTau system that recapitulates the pathological progression seen in human subjects with tau-mediated neurodegeneration in just 21 DIV3D.

## scRNA-seq of AstTau reveals dynamic transcriptional response to oTau-induced pathology

The transcriptome of AstTau was characterized by acquiring robust scRNA-seq profiles for 130,605 single cells from four replicate culture batches with an average of 2000 genes per cell from experimental conditions across a time course of 3 weeks (Supplementary Fig. 2a, b). A comparative analysis of control and AstTau scRNA-seq profiles across 21 DIV3D revealed shared and cell-type-specific transcriptional responses to oTau-induced pathology (Fig. 4). UMAP projection demonstrates that all scRNA-seq experimental condition datasets are integrated, and highlights changes in the relative proportion of cell types between control and AstTau (Fig. 4a, b and Supplementary Fig. 6). At 21 DIV3D there is an expansion of the astrocytic compartment (ASC) in AstTau that correlates with the observed immunolabeling (Fig. 4b). Notably, there is a decrease in the EX_NEU cluster and a correlated increase in the IN_NEU cluster in AstTau consistent with the observation of glutamatergic excitatory neuronal loss in human subjects with tau-mediated neurodegeneration (Fig. 4b and Supplementary Fig. 10d–f).

Differential gene expression (DEG) analysis between the control and AstTau conditions within cell-type clusters identified striking transcriptional changes over the 21 DIV3D time course (Fig. 4c and

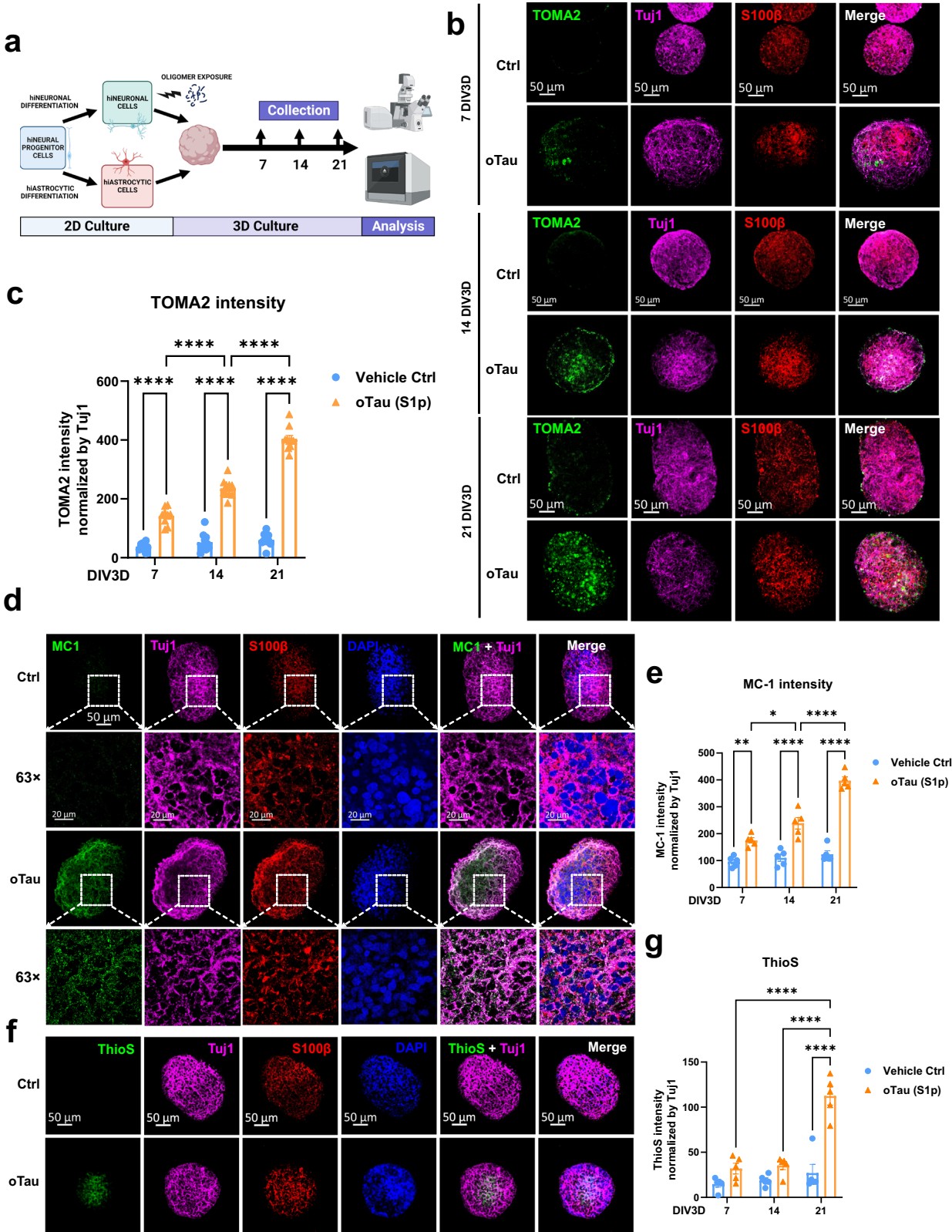

Supplementary Fig. 13). Notably, concurrent transcriptional changes were evident in both the ASC and NEU groups by 7 DIV3D, suggesting a high degree of interaction between the cell types as only NEU populations were exposed to toxic human oligomeric tau (Fig. 4c). AstTau exhibited regulation of ribosomal transcripts, HSP chaperone transcripts, and TNF/MAPK associated intermediate early response (IER) transcripts (termed neuroinflammatory)[69] (Fig. 4c). Unexpectedly, an increase in ribosomal transcripts including *RPS26* and *RPS28* (green) dominated the early response at 7 DIV3D, suggesting a robust translational stress response (Fig. 4c). This response was tested for evidence of a translational stress response by immunolabeling of HNRNPA2B1, an RNA binding protein; HNRNPA2B1 showed abundant cytoplasmic labeling at 7 DIV3D, which is consistent with the presence of a translational stress response (Supplementary Fig. 10g, h). In contrast, there

**Fig. 2 | oTau seeding induces rapid tau misfolding and aggregation in the 3D neuron and astrocyte coculture AstTau model. a** Experimental schematic diagraming 2D culture of hiNCs and hiACs, selective exposure of hiNCs to oTau, 3D asteroid coculture, and immunolabeling and single-cell transcriptomic analysis at indicated data collection timepoints. Created with BioRender.com.
**b** Representative images showing the progressive accumulation of toxic tau oligomers in oTau-seeded asteroids at 7, 14, and 21 DIV3D. Tau oligomers (TOMA2, green), neurons (βIII tubulin (Tuj1), violet), astrocyte (S100β, red). Scale bars = 50 μm. **c** Quantification of oTau by fluorescence intensity of TOMA2, normalized to Tuj1 intensity. Data obtained from ten independent asteroids. Error bars = SEM. Two-way ANOVA with Tukey's multiple comparisons test was performed, ****$P < 0.001$ comparisons to timepoint vehicle control. **d** Representative images

showing tau misfolding in AstTau at 21 DIV3D. Misfolded tau (MC1, green), neurons (Tuj1, violet), astrocyte (S100β, red). The 63× bottom row shows high magnification of the highlighted regions for the oTau panels. Scale bars = 10 mm and 50 μm.
**e** Quantification of MC1 labeled fluorescence intensity normalized to Tuj1 intensity. Data obtained from five independent asteroids. Error bars = SEM. Two-way ANOVA with Tukey's multiple comparisons test was performed, *$P < 0.05$, **$P < 0.01$, ****$P < 0.001$ comparisons to timepoint vehicle control. **f** Representative images showing ThioS labeling at 21 DIV3D. Neurons (Tuj1, violet), astrocytes (S100β, red). Scale bars = 50 μm. **g** Quantification of ThioS-labeled fluorescence intensity normalized to Tuj1 intensity. Data obtained from five independent asteroids. Error bars = SEM. Two-way ANOVA with Tukey's multiple comparisons test was performed, ****$P < 0.001$ comparisons to timepoint vehicle control.

was a suppression of HSP (red) and neuroinflammatory transcripts (purple) at 7 DIV3D (Fig. 4e). This trend was reversed with increasing oTau pathology, and by 21 DIV3D there was an increase in stress response transcripts, including transcripts associated with neuroinflammation (*JUN, DUSP1,* and *RHOB*), and HSP chaperones (*HSPA1B, HSPH1, DNAJB1, HSPB1, HSPD1,* and *HSP90AA1*) (Fig. 4c). Interestingly, EX_NEU demonstrated the most robust upregulation of neuroinflammatory transcripts, while ASC presented with a stronger HSP response (Fig. 4c).

These responses were further elucidated by performing statistical analyses on the average expression of all genes per in select curated MsigDB[70] or literature-derived gene sets per cell to perform a comparative signature analysis (see "Methods") (Fig. 4d, e). This demonstrated the observed transcriptional changes at 21 DIV3D in AstTau ASC align with transcriptional signatures upregulated in classical reactive astrocyte including A1 and A2 reactive astrocyte signatures[71] as well as Alzheimer's Disease (AD)-associated reactive astrocytes, specifically identified in human subjects with AD neurodegeneration by single-nuclei RNA-seq (snRNA-seq)[61] (Fig. 4d). Notably, there was a more significant enrichment ($P < 0.0001$) of the A1 toxic astrocyte signature than the A2 protective astrocyte signature ($P < 0.05$) in AstTau (Fig. 4d). The ASC population also show a significant upregulation of the Reactome HSF1 Activation pathway associated with HSP chaperone production (Fig. 4d).

Comparative signature analysis in the AstTau neuronal populations demonstrated a concurrent transcriptional upregulation with neuronal signatures found in human subjects with AD neurodegeneration by snRNA-seq[72] (Fig. 4e). However, there was divergent regulation of neuroinflammation demonstrated by the HALLMARK TNFα signaling via NFKB signature between IN_NEU and EX_NEU. There was a significant upregulation of neuroinflammation in the EX_NEU population, while the signature was suppressed in the IN_NEU population, demonstrating notable consistency with EX_NEU cell population loss at 21 DIV3D (Fig. 4b and Supplementary Fig. 10d–f). In addition, both EX_NEU and IN_NEU presented a significant downregulation of the GO cytosolic ribosomal pathway at 21 DIV3D potentially indicative of the translational repression that has been associated with tauopathies[73] (Fig. 4e).

Functional gene set enrichment analysis further highlights a trajectory of pathological responses among the AstTau cell types. There was a strong ribosomal pathway enrichment at 7 DIV3D that weakened through 14 and 21 DIV3D in agreement with previous analyses (Fig. 5a, b and Supplementary Fig. 10). Pathways associated with neuroinflammation, neuron stress and senescence, and HSP chaperones were enriched among EX_NEU upregulated genes at 21 DIV3D (Fig. 5a, b and Supplementary Fig. 15). While IN_NEU shared the HSP chaperone pathway enrichment, it did not share the neuroinflammatory pathway enrichment, further supporting cell-type-specific differential responses to pathology progression (Fig. 5a and Supplementary Fig. 15). ASC demonstrated the most robust HSP chaperone pathway enrichment, as well as cell activation, glial development, and stress response pathways enrichment suggestive of reactive astrogliosis (Fig. 5a, b and

Supplementary Fig. 15). Together, the functional gene set pathway enrichment of upregulated transcripts across the 21 DIV3D time course in AstTau reveals a progressive stress response including translational, respirasome, and neuroinflammatory profiles that result in the observed neurodegeneration.

Importantly, functional gene set enrichment analysis also reveals concordant responses between AstTau and single-nuclei transcriptomic profiles of postmortem AD brain, in particular the HSP chaperone response, indicating activation of similar pathophysiological cascades that are not shared by other neuropathologies[61,72,74] (Fig. 5c). These signatures are not shared by comparative controls including single-nuclei transcriptomic profiles of autism spectrum disorder (ASD) brain tissue or scRNA-seq of PFF-αSYN exposed asteroid cultures (AstαSYN) (Fig. 5c), which highlights the specificity of oTau in the development of the pathological and transcriptional evolution observed in AstTau. In addition, mapping of AD-associated GWAS genes revealed modulation in AstTau at 21 DIV3D including an increase in *APOE* expression in ASC with a concurrent suppression in IN_NEU and EX_NEU (Fig. 5d). Interestingly, a subset of AD GWAS genes such as *SORL1* and *CLU* were elevated in AstTau ASC, reduced IN_NEU but unchanged in EX_NEU, which suggests cell-type specificity. Others genes, such as *ADAM17*, presented with the inverse pattern, being elevated in EX_NEU, reduced in IN_NEU but unchanged ASC (Fig. 5d). Finally, *ABCA7* was reduced in each cell type. Together these data point to cell-type specificities in the expression of AD-linked genes.

## Amelioration of AstTau pathology with the epichaperome HSP90 inhibitor PU-H71

The scRNA-seq data profile suggested that the HSP chaperone response plays a pivotal role in neuronal loss and astrocyte activation in our AstTau model. We applied HSP90 inhibitor PU-H71 in our system to test the involvement of HSF1-driven HSP chaperone dysfunction in tau pathogenesis and explore the potential protective effect of correction. Chaperones are proteins that control neuronal function by regulating protein folding, thereby determining the functional levels and activities of many proteins[75,76]. Accumulating intracellular protein aggregates, such as misfolded tau, are thought to compete with functional endogenous proteins for binding to chaperones, creating dysfunctional, high molecular weight tau-chaperone complexes containing HSc70 and 90, rendering neurons prone to degeneration[77,78]. AstTau cultures were treated with HSP90 inhibitor PU-H71 (1 μM, 3 days), which has been shown to modulate the HSF1 transcriptional pathway, stimulating HSP chaperone production (Fig. 6a)[77]. Subsequent analysis of pathology in oTau+/PU-H71+ AstTau showed a striking reduction in tau pathology and a corresponding reduction of neurodegeneration at 21 DIV3D as compared to oTau+/PU-H71 assembloids (Fig. 6b–i). We observed a significant reduction in tau phosphorylation (pS202/5, pS262), misfolding (MC1 antibody), oligomerization (TOMA2 antibody) and tangle formation (ThioS) (Fig. 6b–e). Fluoro Jade B reactivity and LDH release also decreased, indicating a decrease in neuronal cell death (Fig. 6g–i). This reduction in neuronal stress and death was also accompanied by a decrease in

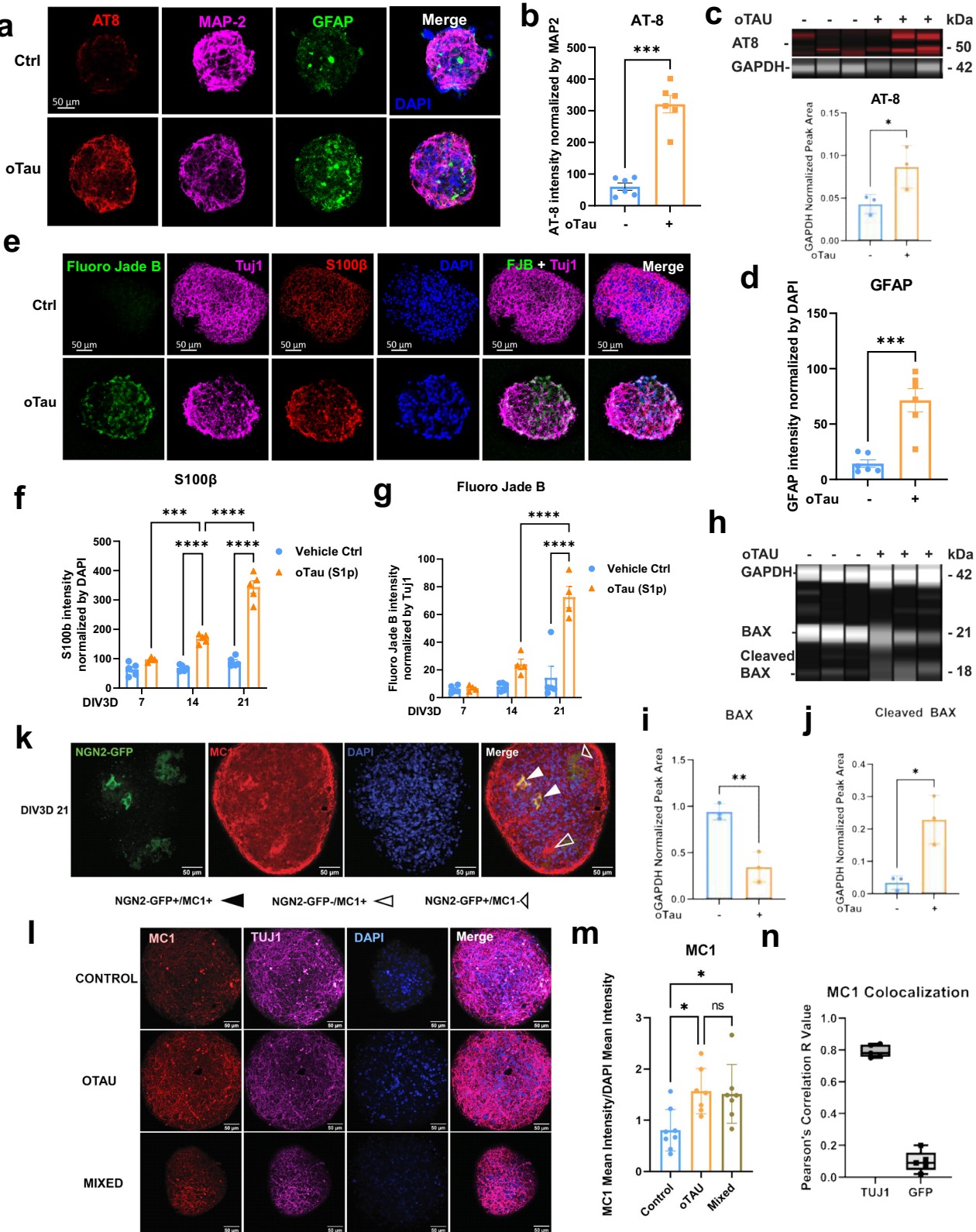

reactive astrogliosis in oTau+/PU-H71+ cultures, marked by decreased S100β and GFAP staining as compared to oTau+/PU-H71− cultures (Fig. 6d, f).

Analysis of the transcriptional responses by scRNA-seq supports the putative benefit observed for PU-H17, with the transcriptional profile shifting to that observed early in the disease process, perhaps prior to a point where the system was overwhelmed by oTau (Fig. 7).

PU-H71+ scRNA-seq cell transcriptomes were integrated with vehicle treated cell transcriptomes and presented the same cell-type clustering and trajectory (Fig. 7a). We observed a reduction in transcriptional changes associated with neuronal cell stress and death, including suppression of the neuroinflammatory transcripts in the EX_NEU population and the HSP chaperone transcripts in IN_NEU (Fig. 7b). Interestingly, PU-H71 initiated a striking re-expression of ribosomal

**Fig. 3 | oTau seeding induces rapid accumulation of toxic tau and neurodegeneration in the AstTau model. a** Representative images of tau hyperphosphorylation (pS202/205 (AT8), red), reactive astrogliosis (GFAP, green), and neurons (βIII tubulin (Tuj1), violet) at 14 DIV3D. Scale bars = 50 μm. **b** Quantification of tau hyperphosphorylation (AT8) normalized to MAP2 fluorescence intensity at 14 DIV3D. Data from five independent asteroids. Error bars = SEM. Two-way ANOVA with Tukey's multiple-comparison test (MCT), ***$P < 0.01$. **c** Immunoblotting and quantification of hyperphosphorylated tau (AT8) normalized to internal control (GAPDH) band intensity. Data from three independent asteroid pools. Error bars = SEM. Unpaired $t$ test, *$P < 0.05$. Source data are provided as a Source Data file. **d** Quantification of reactive astrogliosis (GFAP) normalized to DAPI fluorescence intensity at 14 DIV3D. Data from five independent asteroids. Error bars = SEM. Two-way ANOVA with Tukey's MCT, ****$P < 0.001$. **e** Representative images of neurodegeneration (Fluoro Jade Blue (FJB)), neurons (Tuj1, violet), and astrocytes (S100β, red) at 21 DIV3D. Scale bars = 50 μm. **f** Quantification of astrocytic S100β normalized to DAPI fluorescence intensity. Data from five independent asteroids. Error

bars = SEM. Two-way ANOVA with Tukey's MCT, ***$P < 0.01$, ****$P < 0.001$, comparisons to timepoint vehicle control. **g** Quantification of FJB normalized to Tuj1 fluorescence intensity. Data from five independent asteroids. Error bars = SEM. Two-way ANOVA with Tukey's MCT, ****$P < 0.001$, comparisons to timepoint vehicle control. **h** Immunoblotting of cell death activation by BAX labeling with internal control (GAPDH). **i, j** Quantification of BAX (**i**) and cleaved BAX (**j**) band intensity normalized to GAPDH from (**h**). Data from three independent asteroid pools. Error bars = SEM. Unpaired $t$ test, **$P < 0.01$. **k** Representative image of mixed (NGN2_GFP +/oTau-, NGN2_GFP-/oTau+) cultures evidencing tau propagation (NGN2_GFP +/MC1 +, closed arrows). Misfolded tau (MC1, red), neurons (Tuj1, violet), select neurons NGN2_GFP +. Scale bars = 50 μm. **l** Representative images of propagated misfolded tau (MC1, red) in indicated conditions. Scale bars = 50 μm. **m** Quantification of MC1 normalized by DAPI fluorescence intensity from (**l**). Data from seven independent asteroids. Error bars = SEM. Two-way ANOVA with Tukey's MCT, *$P < 0.05$. **n** Quantification of MC1 colocalization by Pearson's Correlation to Tuj1 and NGN2-GFP in 21 DIV3D mixed cultures.

transcripts at 21 DIV3D similar to that observed at 7 DIV3D in all cell types (Fig. 7b). The ASC population also demonstrated a reduction in the upregulation of the A1 toxic astrocytic signature and an increase in the upregulation of the A2 protective astrocytic signature (Fig. 7c). Neurons no longer demonstrated an upregulation of AD-associated neuronal signatures in the PU-H71+ AstTau assembloids (Fig. 7d). In addition, PU-H71 reversed the upregulation of neuroinflammatory signatures in EX_NEU suggesting a mechanism of protection (Fig. 7d). In these analyses the oTau-/PU-H71+ condition is a useful comparison to assess the cellular response to oTau exposure in the presence of PU-H71 versus the transcriptional alterations caused by PU-H71 itself (Fig. 7). The ASC population demonstrated a slight decrease in intensity of heat shock and gliosis reactions with PU-H71 treatment at 21 DIV3D corresponding with the decreased GFAP immunolabeling, together suggesting an altered reactive astrocyte phenotype (Fig. 7e and Supplementary Fig. 16). Notably, the reversal of late state upregulated signatures was more pronounced in the EX_NEU population than either the ASC or IN_NEU populations, indicating a potentially cell-type-specific targeted treatment effect (Fig. 7e and Supplementary Fig. 16). EnrichmentMap presentation of the 7 DIV3D and 21 DIV3D +/− PU-H71 upregulated functional gene set enrichment pathways in EX_NEU demonstrates the pathway enrichment overlap between 7 DIV3D and 21 DIV3D + PU-H71 that is not shared by the cell stress enriched pathways in the 21 DIV3D − PU-H71 condition (Supplementary Fig. 17). Together these data suggest that targeting the dysfunctional HSF1-driven HSP chaperone system with PU-H71 significantly reduces oTau-induced pathology and neurodegeneration in AstTau, accentuating the involvement of HSPs in tauopathies.

## Pseudotime trajectory subclustering and gene module analysis reveals pathway signatures associated with AstTau neurodegeneration

To expand on our analysis of cell-type-specific responses in AstTau, we performed pseudotime subclustering and trajectory analysis using Monocle3. The Monocle3 single-cell algorithm identifies trajectories of genes expression changes through which cells pass during biological processes such as neurodegeneration. Monocle3 clusters and orders the cells based on the calculated trajectory, producing a branched pseudotime mapping, with each branch representing a cell decision point and differential response[79–83].

As the glutamatergic excitatory neurons (EX_NEU) presented with a unique inflammatory and neurodegenerative profile we first aimed to identify a subset of this population that was closely associated with neurodegenerative pathology. The EX_NEU cluster was isolated from the scRNA-seq dataset and reprocessed using Monocle3 (see "Methods"). Subclustering and pseudotime trajectory inference with the Monocle3 pipeline produced a series of 9 clusters with a trajectory of dynamic gene expression changes progressing largely along the 7–21

DIV3D time course (Fig. 8a−c). However, at later pseudotime stages branching indicative of diverging cell fates emerged (Fig. 8b). Percent composition of the pseudotime clusters revealed an over-representation of AstTau in one of these branched regions, cluster 7 (Fig. 8d). Differential gene expression and module analysis identifying co-regulated differentially expressed genes within each pseudotime cluster revealed that cluster 7 was defined by the genes comprising module 12 (Fig. 8e). Functional gene set enrichment analysis was performed on these genes as before to identify pathway signatures associated with this AstTau-enriched pseudotime cluster. Interestingly, significant signatures included KEGG disease pathways such as Alzheimer's and Parkinson's as well as the previously noted HALLMARK Unfolded Protein Response and MYC Target signatures (Fig. 8g). In addition, other signatures associated with pathological AstTau EX_NEUs included KEGG RNA Degradation and HALLMARK MTORC1 Signaling (Fig. 8g).

The same subclustering analysis pipeline was performed on the ASC cell population using Monocle3, identifying pseudotime cluster 10 that was enriched in late-stage AstTau (Supplementary Fig. 14a−d). The gene set module 42 that defined this cluster of AstTau-associated ASCs was notably headed by *APOE*, which has been strongly implicated in AD glial responses (Supplementary Fig. 14e, f). Functional gene set enrichment analysis showed significant correlation with the KEGG Alzheimer's Disease pathway as well as *APOE-associated* pathways such as HALLMARK Adipogenesis (Supplementary Fig. 14h). The pathological AstTau ASCs also displayed signatures of HALLMARK Apoptosis, Hypoxia, Complement, and the Interferon Gamma Response, adding to the growing body of knowledge that places import on the degenerative and inflammatory contributions of ASCs to the pathological development of tauopathies (Supplementary Fig. 14h). Together these pseudotime trajectory analyses further demonstrate the utility of AstTau as a model of discovery for tauopathies.

## Discussion

We have presented a manipulatable and accessible method for modeling tau-induced degenerative pathology in a human iPSC-derived coculture of neurons and astrocytes in a rapid and reproducible manner. Human oTau seeding in hiNCs before combination with hiACs allows for a discrete analysis of how neuronal tau pathology develops and interacts with neighboring astrocytes in a physiologically relevant spatial context. This targeted and controlled methodology allows for a deeper level of study of tau-induced phenotypes in vitro beyond that previously shown[28–33,84]. Previous studies with microfluidic and organoid models have developed some phenotypes of interest such as tau hyperphosphorylation and neurodegeneration in glutamate toxicity challenged MAPT mutant iPSCs[29,84]. Models such as these and others including 2D models develop tau hyperphosphorylation but no neurodegeneration[12–20], and represent partial models of tauopathy. By

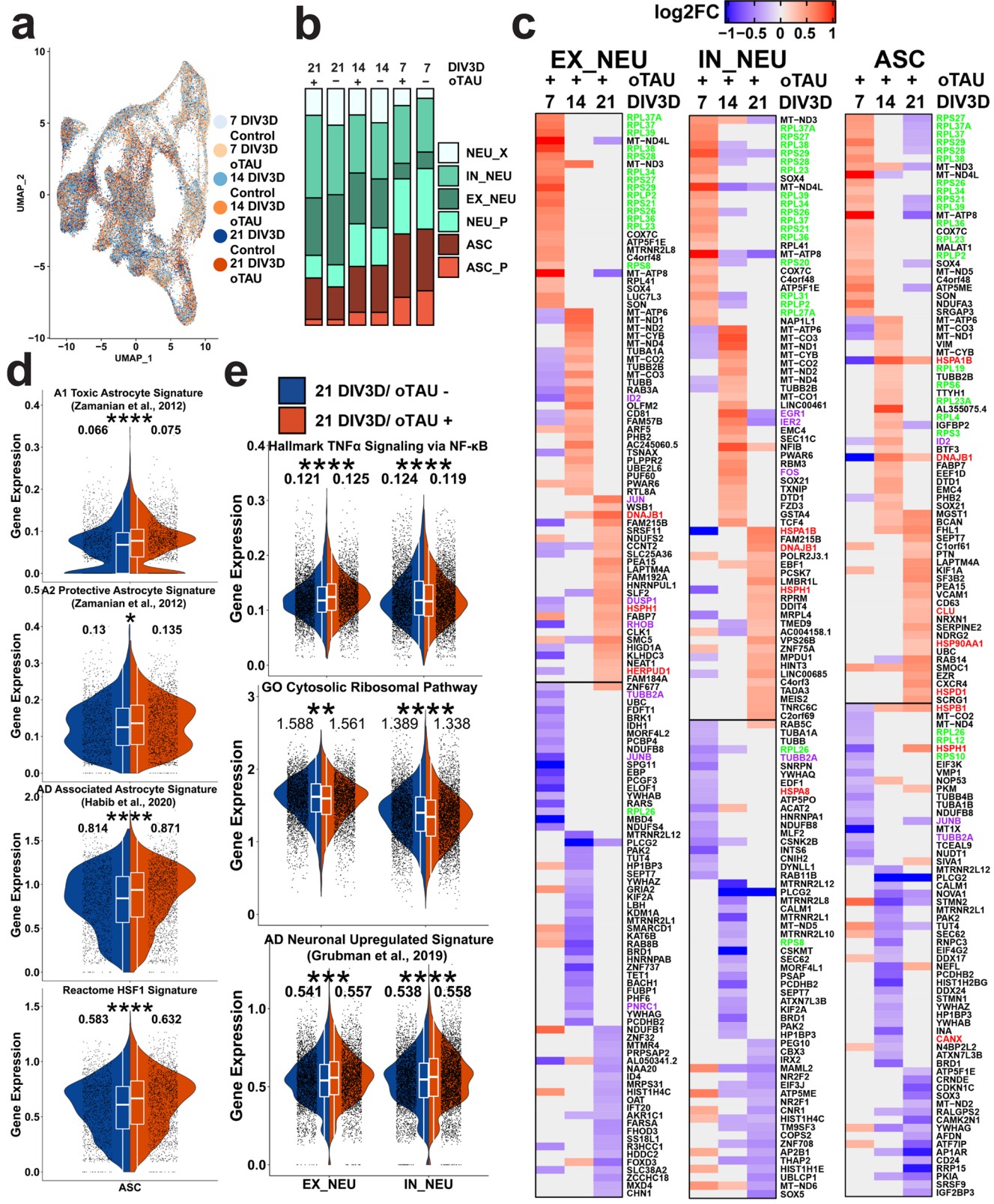

focusing on the oligomeric tau fraction, we were able to induce robust pathology and neurodegeneration over a time course that extends well beyond any acute treatment; this approach also highlights an emerging consensus that oTau is more toxic than fibrillar tau[55,56]. The result is a broader and more clinically relevant model of tau-induced neurodegeneration from oTau seeding and propagation through fibrillar tau formation and neurodegeneration in just 3 weeks.

The AstTau system allows for a detailed analysis of astrocytic phenotypes that are associated with tau pathology in a manner not easily studied with in vivo models that take 6–8 months to develop glial populations[29,85]. The incorporation of hiACs in the AstTau model is necessary for neuronal survival in 3D culture, while also enabling study of the evolution of the interaction over the disease course. AstTau exhibits remarkable astrocytic stress responses resulting in expansion

**Fig. 4 | scRNA-seq of AstTau reveals dynamic transcriptional response to oTau-induced pathology. a** UMAP projection of 130,605 single-cell transcriptomes colored by conditional timepoints across the 7–21 DIV3D control and AstTau time course (blue and orange gradients, respectively) highlighting model integration and trajectory. **b** Percent composition of cell-type clusters in control and AstTau across the 7–21 DIV3D time course indicating loss of EX_NEU and expansion of the ASC compartment in AstTau. **c** Top 25 up- and downregulated DEGs in AstTau by fold change (log2FC) across the 7–21 DIV3D time course in EX_NEU, IN_NEU, and ASC. Transcripts associated with the GO Cytosolic Ribosomal, HALLMARK TNFα Signaling via NFKB, and the REACTOME HSF1 Activation pathways are highlighted in green, purple, and red, respectively. **d, e** Single-cell average gene expression in control (blue) and AstTau (orange) ASC populations at 21 DIV3D highlighting the correlation of AstTau ASC transcriptomic signature with published astrocytic

signatures including A1 toxic and A2 protective astrocytes (ref. [71]), AD-associated astrocytes (ref. [61]), and the Reactome HSF1 pathway (**d**) and EX_NEU and IN_NEU populations at 21 DIV3D highlighting the correlation of AstTau neurons with published single-nuclei AD neuronal signatures (ref. [72]), and the cell-type differential response in the HALLMARK TNFα signaling via NFKB neuroinflammatory response and concurrent response in the GO Cytosolic Ribosomal pathway (**e**). (t.test, ns: $P > 0.05$, $*P <= 0.05$, $**P <= 0.01$, $***P <= 0.001$, $****P <= 0.0001$). Inset box plots show the median, lower and upper hinges that correspond to the first quartile (25th percentile) and third quartile (75th percentile), and the upper and lower whiskers extend from the smallest and largest hinges at most 1.5 times the interquartile range. Mean values are numerically presented. Source data are provided as a Source Data file.

of the astrocytic compartment, selective transcriptional responses, and changes consistent with reactive astrogliosis, including increased GFAP expression. This coordinated stress response is observed as soon as 7 DIV3D, indicating rapid and coordinated crosstalk between neurons exposed to tau oligomers and the otherwise healthy astrocytes. These results highlight the developing importance of glial cell types such as astrocytes to the pathogenesis of neurodegenerative diseases, where they have been found to play roles in inflammation and cell death mediation[57,58]. The reactive astrogliosis in AstTau resembles astrocytic responses observed in AD brain tissue and a subset of pathology-associated astrocytes are enriched for the AD risk factor *APOE*, highlighting the ability of the AstTau system to reproduce physiologically relevant cell-type responses[58].

Importantly, we have demonstrated how the observed pathologies are selective for oTau as shown by the absence of such pathology observed in immunodepleted oTau and PFF-αSYN-treated controls. Neither tau pathology nor neurodegeneration developed upon treatment with immunodepleted oTau, nor did it occur in neurons or astrocytes treated with PFF-αSYN. Despite the lack of toxic response, PFF-αSYN did induce wide-spread phosphorylated α-synuclein that was evident by immunolabeling. Together these data suggest that the particular degenerative tauopathy observed in AstTau is selective for the type of protein aggregate.

This study provides a single-cell transcriptomic analysis of a 3D coculture-based seeded tauopathy model. Informatic analysis identified astrocytic and neuronal cell populations in the AstTau model, and we were able to define inhibitory GABAergic and excitatory glutamatergic neuronal subpopulations produced by a selective NGN2 over-expression differentiation system[86]. These neuronal populations showed distinct responses to toxic oTau exposure, recapitulating phenotypes observed in postmortem AD brain tissue[60,72,74,87]. There was a loss of glutamatergic neurons in AstTau, consistent with reports that show excitatory neurons are more susceptible to neurodegeneration in tauopathies[88]. Differential gene expression revealed that the glutamatergic neurons mounted a more robust late-stage neuroinflammatory response than GABAergic neurons, offering insight to the mechanism of cell-type-specific degeneration. Pseudotime trajectory analysis identified a subset of pathology-associated glutamatergic neurons expressing functional gene set enrichment signatures including the unfolded protein response and RNA degradation, revealing putative tauopathy-associated biological processes at high resolution. In addition, the astrocytic transcriptomic response to neuronal injury was surprisingly rapid and synchronized with the neuronal responses. Astrocytes and neurons both mounted early robust ribosomal responses, consistent with an adaptation to stress[89–94]. Astrocytes were notable for mounting a more robust late-stage HSP response as well as presenting with upregulation of autophagic and astrogliosis pathways, perhaps contributing to their underlying resilience and enabling regulation of neuronal response[95,96]. These AstTau-specific reactive astrocytes showed remarkable transcriptomic similarity to AD-specific disease-associated astrocytes

(DAAs) identified by Habib et al.[61]. In addition, the transcriptomic signature correlates more significantly with the neurotoxic A1 astrocyte signature than the A2 neuroprotective signature, as defined in ref. [71], supporting a potential detrimental role for reactive astrocytes in tauopathy.

Our studies suggest that disruption of chaperone function mediated by tau oligomers contributes to the pathophysiology of tauopathies. Previous work has noted the potential importance of the HSP chaperone system in neurodegeneration[76,97–103]. The HSP chaperone response in AstTau was shared with recent single-nuclei postmortem late-stage AD transcriptomic studies[72,74]. Based on this, we hypothesized that pharmacological modulation by HSP90 inhibitor PU-H71 would validate the involvement of HSF1-driven HSP chaperones in AstTau pathogenesis. This hypothesis is supported by recent work that suggests a dysfunctional HSP chaperone system in disease can be ameliorated by PU-H71 through the production of more functional molecular chaperones (HSc70 and HSP90) as shown in 2D iPSC and mouse models of tauopathy[77,78,104]. We have now shown striking protection by PU-H71 treatment in the iPSC AstTau system that includes rescuing neurons from degeneration. PU-H71 treatment normalized neuroinflammatory transcriptional signatures, as well as reactivated the ribosomal stress response, which suggest that this response can be beneficial. It is notable that this was a short treatment with PU-H71 late in the disease process provided benefit, raising the possibility that HSF1 activation could be beneficial for therapy of tauopathies later in the disease course. These types of findings point to the utility of the AstTau system as a translational tool for drug development.

AstTau was developed to begin to address the need for modeling the contribution of toxic tau oligomers to the pathogenesis of tauopathies in a physiologically relevant human context. The use of iPSC-derived neurons approaches this goal, however as with all iPSC neurons the cells express mostly 3R tau. This challenge remains to be solved. In addition, the model is currently limited in the range of glial cells represented, but future studies will continue to expand the cell-type composition and interactions by including microglia, oligodendrocytes, and potentially vascular cell types. In addition, while this study focuses on the specific impact of toxic tau oligomers in tauopathy development in healthy iPSC-derived cells, the AstTau model is well-suited to be used with a range of iPSC backgrounds including those with mutations that have been shown to contribute to disease progression. Future studies with AstTau could model dementia subtypes by propagating varying oligomeric tau conformers in neurons or astrocytes carrying specific disease-associated genetic mutations. These strengths of the rapid and reproducible AstTau model evidence a powerful and accessible tool for the study of tauopathies.

## Methods
### Compliance statement
All research was performed with approval from Boston University and Boston University School of Medicine (IRBs 18-2277 and 17-785). The use of animals for the derivation of the oligomeric S1p tau

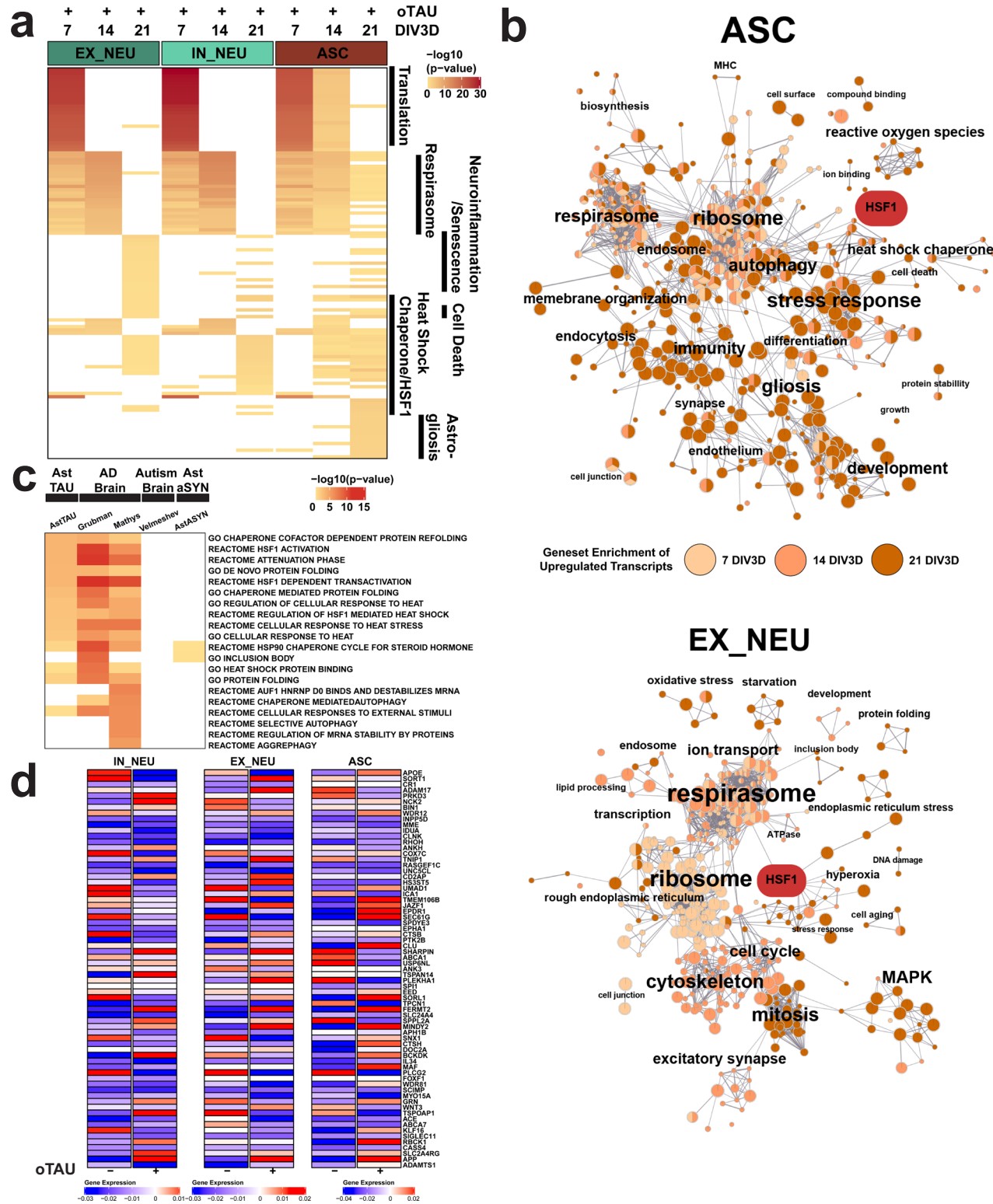

fraction was approved by the Boston University Institutional and Animal Care and Use Committee (AN15301, PROTO201800234). All animals were housed in IACUC-approved vivariums at Boston University School of Medicine.

**Oligomer processing**

**Generation of S1p oTau fraction.** PS19 P301S tau transgenic mice overexpressing human P301S Tau (B6;C3-Tg(Prnp-MAPT*P301S) PS19Vle/J, stock #008169) were purchased from Jackson Laboratories.

Male and female PS19 P301S tau$^{+/-}$ mice were used as breeding pairs and the F1 generation of P301S tau$^{+/-}$ (PS19) and P301S tau$^{-/-}$ (wild type) were used for the experiment. Frozen hippocampus and cortical tissues of 9-month-old PS19 mice were weighed (100–250 mg) and put in Beckman Centrifuge Tubes, polycarbonate thick wall (cat # 362305). A 10× volume of homogenization buffer, Hsaio TBS buffer (50 mM Tris, pH 8.0, 274 mM NaCl, 5 mM KCl) supplemented with protease and phosphatase inhibitor cocktails (Roche, cat# 05892791001 and cat# 04906837001), was used to homogenize brain tissue[55,66]. Briefly, the

**Fig. 5 | Functional gene set enrichment of scRNA-seq DEGs reveals oTau-induced pathway perturbations in AstTau. a** Top 25 functional gene set enrichment pathways by *P* value of significant upregulated DEGs (see "Methods", *P* < 0.05, log2foldchange > 0.25) between AstTau and control for each timepoint in EX_NEU, IN_NEU, and ASC cell populations presented as −log10 *P* value highlighting trends of cell-type-specific pathway perturbations across the 7–21 DIV3D time course. Manual annotation of shared gene set features presented for clarity. See Supplementary Fig. 10 for full annotations. **b** EnrichmentMap clustering presentation of functional gene set enrichment results (see "Methods", *P* < 0.05) with manual annotations across the 7 (light orange), 14 (orange), and 21 DIV3D (dark orange) time course for significantly upregulated DEGs in EX_NEU and ASC, identifying a wide range of cell-type-specific upregulated responses in AstTau. An HSF1-associated cluster is highlighted in red. **c** Top ten functional gene set enrichment pathways from the 21 DIV3D AstTau model and two published AD snRNA-seq datasets (refs. 72 and 74) presented as −log10 *P* value alongside functional gene set enrichment results from a published autism snRNA-seq dataset (ref. 116) and the 21 DIV3D AstαSYN model highlighting the enrichment similarity between AstTau and AD brain transcriptomes and the disparity between AstTau and AstαSYN and ASD brain transcriptomes (see "Methods"). Source data are provided as a Source Data file. **d** Scaled gene expression of AD GWAS genes[124] in IN_NEU, EX_NEU, and ASC at 21 DIV3D control (oTau−) and AstTau (oTau+) conditions.

homogenate was centrifuged at 48,300 × *g* for 20 min at 4 °C. The supernatant was then centrifuged a second time at 186,340 × *g* at 4 °C for 40 min. The TBS-extractable pellet (S1p) fraction was resuspended in a 4× volume of TE buffer relative to the starting weight of the tissue homogenate, aliquoted, and frozen at −80 °C.

**S1p oTau fraction quantification.** The molecular weight of tau in the S1p fractions was documented by native page gel electrophoresis, and the concentration of total tau were measured by immunoblot using 3–12% reducing SDS-PAGE gel by comparison to gradient concentrations of recombinant tau ladders, using the human-specific tau13 antibody (detecting total human tau) by immunoblot (Supplementary Fig. 4)[55] All the fractions were then normalized and divided into fractions of 20 μg/ml tau for storage and future use.

**Immunodepletion of tau from S1p oTau fraction.** Tau aggregates in S1p oTau fractions were eliminated by a direct immunoprecipitation kit (Pierce, cat# 26148). Briefly, the tau-5 antibody was coupled to AminoLink plus Coupling Resin, and the fractions were pre-cleared using the Control Agarose Resin with all the materials provided by the kit. The sample was added to the antibody-coupled resin in the spin column and incubated in the column overnight at 4 °C on a gentle rotator. The column was centrifuged, and the flow-through was saved for further experimentation. After three washes with IP buffer, the spin column was placed into a new collection tube, and tau plus antibodies were eluted from the resin. The eluate was analyzed for the presence of tau.

**FITC labeling of S1p oTau fraction.** The DyLight® 488 Conjugation Kit / DyLight® 488 Labeling Kit (Abcam, cat#ab201799) was used for a simple and quick process for DyLight® 488 labeling/conjugation of S1p oTau fraction. Detailed steps for conjugation of the S1p oTau fraction were followed to the manufacturer's instruction. Briefly, a modifier was added to S1p oTau fraction aliquot and incubated for 15 min followed by adding quencher and incubation for 5 min. The DyLight® 488-conjugated oTau fraction can be used immediately in cell culture treatment.

**Generation and quantification of PFF-αSYN.** Preformed fibrillar α-synuclein (PFF-αSYN) was produced following standardized methodology[105]. Human α-synuclein monomers (Proteos cat# RP-003) were thawed and centrifuged at 15,000×*g* for 10 min at 4 °C. The supernatant was transferred to a 1.5-mL microcentrifuge tube, and the protein concentration was measured by 280 nM absorbance spectrophotometry. Protein concentration was then adjusted to 5 mg/mL with 1× dPBS, and monomers were agitated in an orbital thermomixer with a heated lid at 1000 RPM for 7 days at 37 °C. Fibrils were quantified by a Thioflavin T assay and transmission electron microscopy.

## Cell culture and treatment
All cell cultures were maintained at 37 °C with 5% CO₂. All cell counts were performed in quadruplicate using the Cellometer K2 with AOPI viability dye (Nexcelom) or the Countess3 with 0.4% trypan blue dye (Thermo Fisher).

**Neural progenitor cell (NPC) culture.** Human iPSC (XCL-1) derived neural progenitor cells (NPCs, Stem Cell Tech cat#70901) were maintained in serum-free STEMdiff™ Neural Progenitor Medium 2 (Stem Cell Tech cat#08560) on Corning® Matrigel® hESC-qualified Matrix (Corning cat#354277) coated tissue culture plates. NPCs were plated at 50,000 cells/cm² and passaged at 90% confluency by Accutase™ (Stem Cell Tech cat#07920) dissociation as necessary. A full media change was performed every other day. Low passage (passage < 3) NPCs were cryopreserved in STEMdiff™ Neural Progenitor Medium 2 with 10% DMSO, and all NPCs used for experimentation were maintained at passage <6.

**iPSC neuronal cell (hiNC) differentiation.** NPCs were passaged and plated at 50,000 cells/cm² in STEMdiff™ Forebrain Neuron Differentiation Media (Stem Cell Tech cat#08600) on Corning® Matrigel® coated tissue culture treated plates and transduced with a NEUROG2 lentivirus (GeneCopoeia cat#LPP-T7381-Lv105-A00-S) at MOI 3 to induce iPSC-derived neuronal cells (hiNC). After 24 h of transduction, a full media change was performed.

**iPSC astrocytic cell (hiAC) differentiation.** iPSC-derived astrocytic cells (hiAC) were differentiated from NPCs by small molecule differentiation in STEMdiff™ Astrocyte Differentiation Media (Stem Cell Tech #cat100-0013) on Corning® Matrigel®-coated tissue culture treated plates. A full media change was performed daily for 4 days, and NPC cultures were passaged at 90% confluence by Accutase™. NPCs were reseeded at 150,000 cell/cm² and culture was continued in STEMdiff™ Astrocyte Differentiation Media with a full media change every other day for 14 days, passaging as necessary with Accutase™. At this stage the differentiated Astrocyte Precursor Cells (APCs) were cryopreserved in STEMdiff™ Astrocyte Differentiation Media with 10% DMSO. At the time of experimentation, APCs were thawed and plated at 150,000 cell/cm² in STEMdiff™ Astrocyte Maturation Media (Stem Cell Tech cat# 100-0016) on Corning® Matrigel® coated tissue culture treated plate. A full media change was performed every other day for 6 days, with one passage by Accutase™ at 90% confluence as necessary.

**Asteroid generation and maintenance.** A single-cell suspension of hiNCs and hiACs was prepared by Accutase™ dissociation and washed once with DMEM/F12 (Stem Cell Tech cat# 36254) to remove debris. hiNCs and hiACs were combined at a 1:1 ratio in Asteroid Media (DMEM/F12 (Stem Cell Tech cat# 36254), 1% Glutamax (Thermo Scientific cat# 35050061), 1% sodium pyruvate (Thermo Scientific cat# 11360070), 1% N-2 Supplement (Thermo Scientific cat# 17502-048), 1% B-27 Supplement (Thermo Scientific cat# 17504044), 10 μM Y-27632 (EMD Millipore cat# SCM075) 1% PenStrep (Thermo Scientific cat# 15140148), 1 mg/mL Heparin (Sigma-Aldrich cat# H3149-250KU)) and plated in AggreWell™800 microwells (Stem Cell Tech cat# 34815) coated with Anti-Adherence Rinsing Solution (Stem Cell Tech

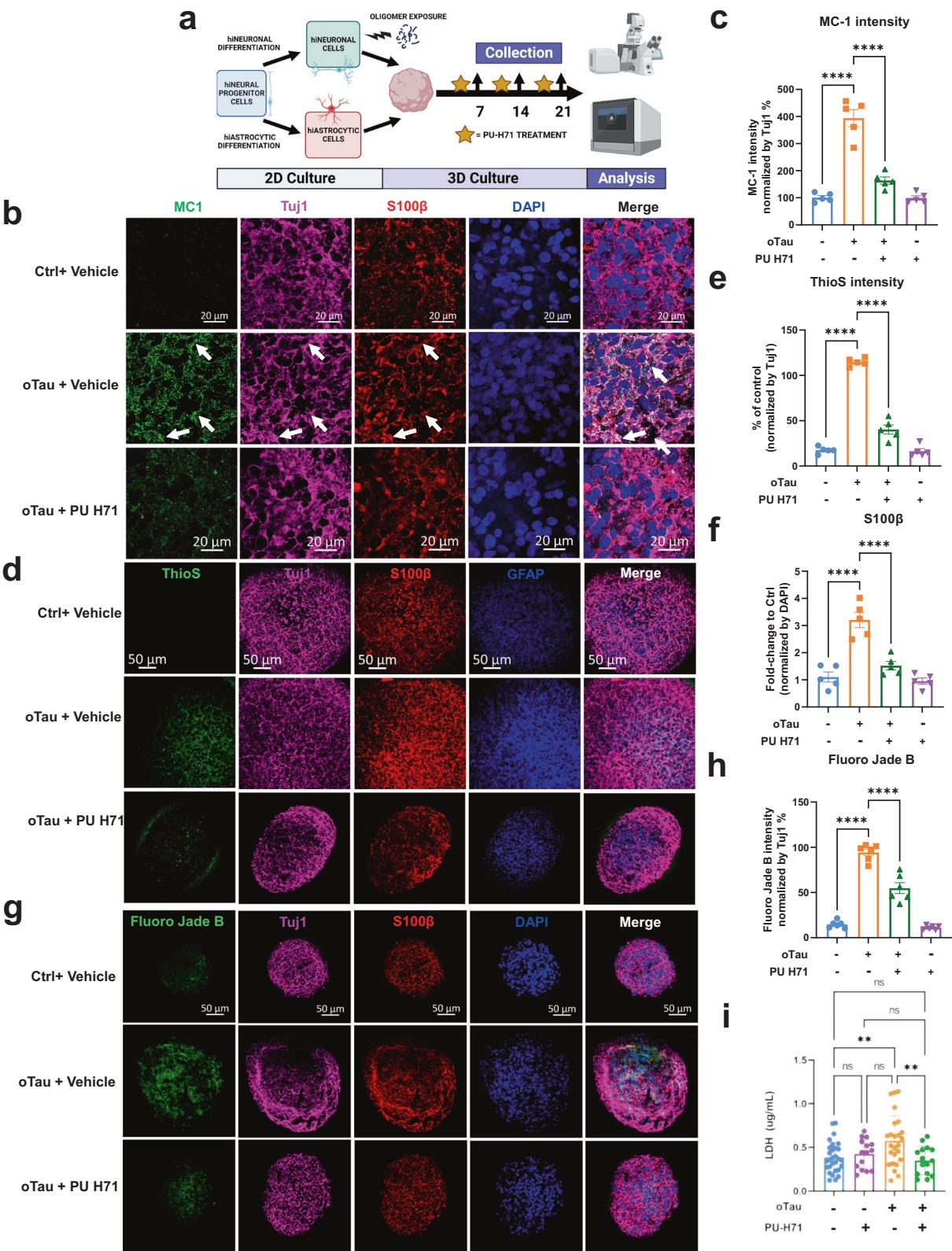

cat#07010). The Aggrewell™ plate was immediately centrifuged at 100×*g* for 3 min to capture the cells in the microwells and incubated for 24 h. A half-media change was performed at 24 h and then every other day for 1 week. At 1 week when the spheroids displayed a smooth, bright edge under the cell culture microscope cultures were transferred to ultra-low-attachment round-bottom 96-well plates (Fisher Scientific cat#07-201-680) and maintained in 100–200 μL asteroid

media rotating at 85 rpm. A half-media change was performed every other day for up to 3 weeks. Cultures were monitored, and live images were captured using the AmScope or EVOS m7000 platforms.

**hiNC oTau treatment.** hiNCs were selectively exposed to 0.04 mg/mL oTau by direct administration in cell culture media for 24 h before incorporation into asteroid culture.

**Fig. 6 | PU-H71 treatment reduces oTau-induced pathology and neurodegeneration. a** Experimental schematic including 2D culture, oligomer exposure, 3D culture, PU-H71 treatment, and analysis at indicated timepoints. Created with BioRender.com. **b** Representative images showing reduction in tau misfolding in AstTau upon exposure to PU-H71 for 3 days, from 19 to 21 DIV3D. Misfolded tau (MC1, green), neurons (βIII tubulin (Tuj1), violet), astrocyte (S100β, red). Scale bars = 20 μm. **c** Quantification of MC1 labeled fluorescence intensity normalized to Tuj1 intensity. Data obtained from 5 independent asteroids. Error bars = SEM. Two-way ANOVA with Tukey's multiple comparisons test was performed, ****$P < 0.0001$ comparisons to timepoint vehicle control. **d** Representative images showing reduction in ThioS (fibrillar tau), in AstTau upon exposure to PU-H71 for 3 days, from 19 to 21 DIV3D. Misfolded tau (MC1, green), neurons (Tuj1, violet), astrocyte (S100β, red) and (GFAP, blue). Scale bars =50 μm. **e, f** Quantification of ThioS (**e**) and

S100B (**f**) labeled fluorescence intensity normalized to Tuj1 (**e**) or DAPI (**f**) intensity. Data obtained from five independent asteroids. Error bars = SEM. Two-way ANOVA with Tukey's multiple comparisons test was performed, ****$P < 0.0001$ comparisons to vehicle control at each timepoint. **g** Representative images showing Fluoro Jade B-labeled fluorescence in AstTau upon exposure to PU-H71 for 3 days, from 19 to 21 DIV3D. Damaged neurons are labeled by Fluoro Jade B (green). Neurons (Tuj1, violet), astrocyte (S100β, red) and (GFAP, blue). Scale bars = 50 μm. **h** Quantification of Fluoro Jade B-labeled fluorescence intensity normalized to Tuj1 intensity. Data obtained from five independent asteroids. Error bars = SEM. Two-way ANOVA with Tukey's multiple comparisons test was performed, ****$P < 0.0001$ comparisons to vehicle control at each timepoint. **i** LDH detection in the conditioned medium of 21 DIV3D asteroids. Data obtained from 25 independent asteroids. Error bars = SEM. One-way ANOVA with Tukey's multiple comparisons test was performed, **$P < 0.01$.

**hiNC PFF-αSYN treatment.** Sonication of prepared fibrillar α-synuclein was performed to produce preformed fibrillar α-synuclein (PFF-αSYN) for use in experimentation. Briefly, fibrils were diluted in 1× dPBS to 4 μg/μL and sonicated with a 3.2 mm diameter probe for 1 min of 60 pulses at an amplitude of 30%. hiNCs were selectively exposed to 1 μg/mL PFF-αSYN by direct administration in cell culture media for 24 h before incorporation into asteroid culture.

**PU-H71 treatment.** Asteroids were treated with 1 μM PU-H71 by direct administration in cell culture media for 72 h before timepoint collection.

**Mixed culture.** To determine the extent of seeded oTau propagation in the AstTau model, the following experiment was performed. Two sets of asteroid cultures were established, NGN2_GFP+/oTau− and NGN2_GFP-/oTau+. NGN2_GFP (NEUROG2) lentivirus was transduced at MOI 3 (GeneCopoeia cat#LPP-T7381-Lv103-A00-S) as described in "iPSC neuronal cell (hiNC) differentiation". At the 1-week timepoint NGN2_GFP +/oTau - and NGN2_GFP-/oTau + asteroids were mixed and placed in 96-well culture as described in "Asteroid generation and maintenance". The asteroids merged together and were cultured to 21 DIV3D.

**Sample collection**
**Asteroid fixation.** At the time of collection, asteroids were transferred to a 1.5-mL Protein LoBind Eppendorf (Eppendorf cat#022-43-108-1) and allowed to settle. The supernatant was discarded, and asteroids were fixed in 4 °C 4% PFA in 1× PBS for 15 min, rotating at room temperature. After fixation, asteroids were washed 3× for 10 min each with 4 °C 1× PBS, rotating at room temperature. Samples were stored in 1× PBS at 4 °C.

**Protein lysate.** At the time of collection 50 pooled asteroids were transferred to a 1.5 mL Protein LoBind Eppendorf (Eppendorf cat#022-43-108-1) and allowed to settle. The supernatant was discarded, and samples were flash-frozen and stored at −80 °C. Samples were subsequently lysed by mechanical homogenization in 50 μL RIPA lysis buffer (150 mM NaCl, 50 mM Tris-HCL pH 7.4, 1 mM EDTA pH 8.0, 1% NP-40, 0.25% sodium deoxycholate) with Halt™ Protease Inhibitor Cocktail (Thermo Scientific cat#78430), cOmplete™ Protease Inhibitor Cocktail (Roche cat#11697498001), and PhosSTOP™ (Sigma-Aldrich cat#4906845001) for 40 s on ice. Lysate concentration was quantified by the Pierce BCA assay according to the manufacturer's protocol (Thermo Fisher cat#23225).

**Conditioned media collection.** In all, 50 μL of conditioned cell culture media from replicate asteroids were collected in flat-bottomed 96-well plates and frozen at −20 °C.

**Preparation of asteroids for electrophysiological recordings.** Asteroids from each condition (1–4 asteroids per well) were embedded within Geltrex (Thermo Fisher cat# A1413301) in a 48-well plate to

ensure stability during electrophysiological recording. Embedded asteroids were then equilibrated in oxygenated (aerated with 95% $O_2$, 5% $CO_2$) Ringer's solution (concentrations in mM: 126 NaCl, 2.5 KCl, 1.25 $NaH_2PO_4$, 2 $MgCl_2$, 2 $CaCl_2$, 26 $NaHCO_3$, and 10 glucose, 311 mOsm, Sigma-Aldrich) at 34 °C for at least one hour before recording. After equilibration, embedded asteroids were placed into a submersion-type recording chamber (Harvard Apparatus, Holliston, MA, USA) mounted on the stage of a customized uMs-Nikon FN1 infrared-differential interference contrast (IR-DIC) microscope (Sensapex, Oulu, Finland) and were continuously superfused (2–2.5 mL/min) with oxygenated Ringer's solution maintained at 34 °C with an in-line solution heater (35 C Warner Instruments).

**Sample processing**
**Immunofluorescence labeling.** For immunolabeling, selected asteroids from each condition were washed in 150 μl PBS for 10 min in a U-bottom 96-well plate and then permeabilized in 150 μl PBS/0.01% Triton X-100 (PBST). The asteroids were then blocked in PBST supplemented with 5% BSA and 5% normal donkey serum for 1.5–2 h at room temperature (RT). After blocking, asteroids were incubated in primary antibodies diluted in 5% BSA/PBST overnight at 4 °C. On the second day, the asteroids were washed three times in PBST, 15 min each, before they were transferred into 2° antibodies dilute (1:700 of Dylight-/Alexa-conjugated antibodies made in donkey purchased from Thermo Fisher Scientific in 5% BSA/PBST) for 2 h at RT. For DAPI nuclei stain, DAPI (1:10,000) was diluted in PBST and incubated with asteroids for 15 min, followed by 2× washes with PBST then 1× with PBS, 10 min each. The asteroids were then mounted onto microscope glass slides in Prolong gold antifade reagent (Thermo Fisher cat#P36930) and stored in the dark until imaging. Primary antibodies used for asteroid labeling were as follows: Tuj1/βIII-Tubulin (chicken, SYSY, cat# 302 306, 1: 300), MAP2 (rabbit, Millipore, cat# AB5622, 1:1000), Rabbit monoclonal anti-S100β (Abcam, Cat# ab52642, 1:400); GFAP Monoclonal Antibody (Thermo Fisher Scientific, Cat#13-0300, 1:400); Mouse monoclonal anti-TOMA2 (provided by Dr. Rakez Kayed), 1:300; MC1 (provided by Dr. Peter Davies, Northwell), 1:300; CP13 (provided by Peter Davies, Northwell), 1:300; AT8 (Thermo Fisher, cat# MN1020), 1:300, HNRNPA2B1 (Thermo Fisher Scientific, cat# PA534939, 1:500), VGLUT1 (Synaptic Systems, Cat#135-203, 1:100), GAD67 (Thermo Fisher Scientific, Cat#PA5-19065, 1:100).

**Thioflavin S staining.** Fresh Thioflavin S (ThioS) solution was prepared by dissolving 1 g of ThioS (Millipore Sigma cat#T1892) in 100 ml 80% ethanol, and stirring overnight at 4 °C, and filtering for final use. The asteroids to be stained were washed sequentially in 70% and 80% ethanol, 1 min each, prior to incubating in ThioS/80% ethanol solution for 15 min. Asteroids were then sequentially washed in 80% and 70% ethanol, 1 min each, followed by two rinses in PBS. Asteroids were mounted in Prolong Gold antifade reagent (Thermo Fisher cat#P36930) and stored in the dark until imaging.

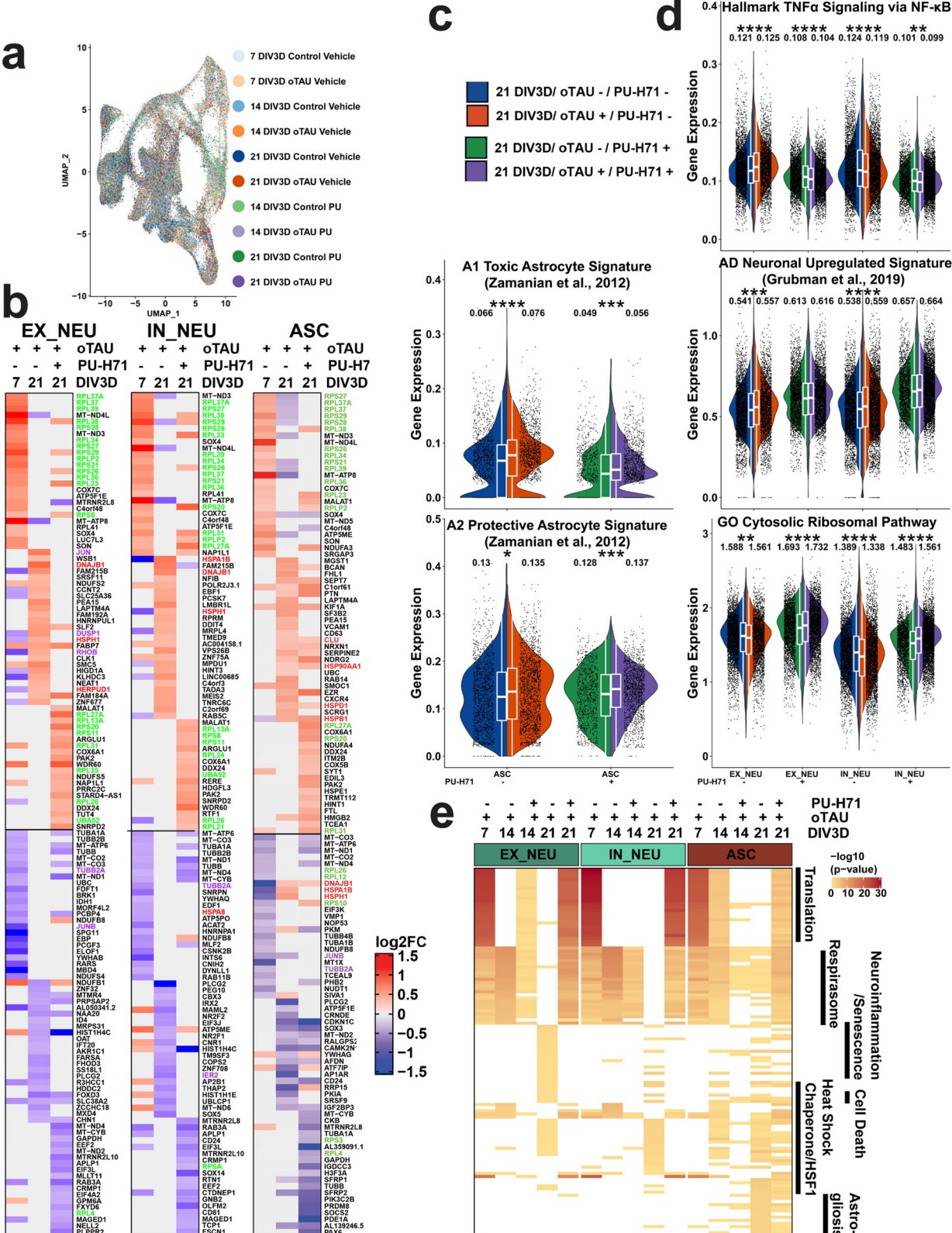

**Fluoro Jade B staining.** The Fluoro Jade B reagent was purchased from EMD Millipore (Cat# AG310-30MG) and the staining protocol was followed as instructed by the manufacturer. Briefly, the staining solution was prepared from a 0.01% stock solution for Fluoro Jade B that was made by adding 10 mg of the dye powder to 100 mL of distilled water. To make up 100 mL of staining solution, 4 mL of the stock solution was added to 96 mL of 0.1% acetic acid vehicle. This results in a final dye concentration of 0.0004%. The stock solution, when stored in the refrigerator, was stable for months, whereas the staining solution was typically prepared within 10 min of use and was not reused. Before staining, the asteroids were rinsed in distilled water and were then treated with 0.06% $KMnO_4$ solution for 15 min. Then the asteroids were stained with Fluoro Jade B working solution for 30 min followed by being washed with PBS twice for 5 min each. Asteroids were mounted

**Fig. 7 | scRNA-seq transcriptomic modulation supports PU-H71 reversal of the pathological molecular phenotype in AstTau. a** UMAP projection of 130,605 cells colored by conditional timepoints across the 7–21 DIV3D control and AstTau time course with vehicle or PU-H71 treatment (blue, orange, green, purple gradients, respectively) highlighting model integration and trajectory. **b** Top 25 up and downregulated DEGs between control and AstTau at 7 DIV3D and 21 DIV3D +/− PU-H71 treatment in EX_NEU, IN_NEU, and ASC, highlighting the similarity between 7 DIV3D and 21 DIV3D with PU-H71. Transcripts associated with the GO Cytosolic Ribosomal, HALLMARK TNFα Signaling via NFKB, and the REACTOME HSF1 Activation pathways are highlighted in green, purple, and red, respectively. **c, d** Single-cell average gene expression in the oTau-/PU-H71- (blue), oTau+/PU-H71-(orange), oTau-/PU-H71 + (green), and oTau+/PU-H71 + (purple) ASC populations (**c**) highlighting the impact of PU-H71 treatment on the A1 toxic and A2 protective astrocytic signature[71], and NEU populations (**d**) highlighting the impact of PU-H71 treatment

on the neuronal signatures including single-nuclei AD neuronal signatures (ref. [72]), the HALLMARK TNFα signaling via NFKB neuroinflammatory response, and the GO Cytosolic Ribosomal at 21 DIV3D. (t.test, ns: $P > 0.05$, *$P < = 0.05$, **$P < = 0.01$, ***$P < = 0.001$, ****$P < = 0.0001$). Inset box plots show the median, lower and upper hinges that correspond to the first quartile (25th percentile) and third quartile (75th percentile), and the upper and lower whiskers extend from the smallest and largest hinges at most 1.5 times the interquartile range. Mean values are numerically presented. **e** Top 25 functional gene set enrichment pathways by $P$ value of significant upregulated DEGs (see "Methods", $P < 0.05$, log2foldchange > 0.25) between Ast-Tau and control for each timepoint in EX_NEU, IN_NEU, and ASC cell populations +/− PU-H71 presented as −log10 $P$ value highlighting cell-type-specific pathway perturbations that are ameliorated by PU-H71 treatment. Manual annotation of shared gene set features presented for clarity. See Supplementary Fig. 10 for full annotations. Source data are provided as a Source Data file.

---

in Prolong Gold antifade reagent (Thermo Fisher cat#P36930) and stored in the dark until imaging.

**Immunofluorescence labeling of 10-µm thin sections.** A pool of ten asteroids per condition were fixed with a fresh 4% paraformaldehyde (PFA) and 0.1% glutaraldehyde solution in a 1.5-ml tube for 25 min. After fixation, the asteroids were washed 3× for 10 min each with PBS. The samples were stored in PBS with 0.025% sodium azide at 2–8 °C for up to 1 week. 24 h before sectioning, tissue was equilibrated in a 30% sucrose solution to protect against freezing artifacts. On the day of sectioning, the tissue was embedded in 200 µL gelatin solution (7.5% gelatin and 10% sucrose in PBS) for 1 h at 37 °C. We then removed the tissue gelation bud from 1.5-ml tube and mounted it on a cryostat embedding mold. 10 µm thin sections were collected directly to positive-charged glass slides. Sections were stored at −20 °C. For immunofluorescence labeling, sectioned slides were removed from freezer and allowed to dry at room temperature. Sections were outlined with a PAP pen, and once dry we proceeded with our regular immunofluorescence labeling protocol as described in "immunofluorescence labeling". The high-magnification images of the asteroid thin sections were captured by confocal Zeiss LSM 880 with Airyscan mode.

**Immunoblotting.** In all, 1 µg total protein lysates were analyzed on the ProteinSimple SimpleWestern Jess platform with the 12–230 kDA microplate kit (ProteinSimple cat#SM-W004) according to the manufacturer's instructions. Primary antibodies were detected using multiplexed chemiluminescent and fluorescent detection modules according to the manufacturer's instructions (ProteinSimple cat#sDM-001, DM-002, DM-007, DM-008, DM-009, DM-010). Primary antibodies used for immunolabeling were as follows: GAPDH (mouse, ProteinTech, 60040-I-Ig, 1: 500), BAX (mouse, Santa Cruz, sc-7480, 1: 25), AT8 p202 Tau (mouse, Thermo Fisher, MN1020, 1:50).

**LDH cytotoxicity assay.** The CytoTox 96 Non-Radioactive Cytotoxicity Assay or LDH-Glo Cytotoxicity Assay was performed as per manufacturer's instructions using 50 µL conditioned media replicates to measure lactate dehydrogenase (LDH) release (Promega cat# G1780, J2380). For the CytoTox 96 assay 490 nm absorbance readings were taken on a SpectraMaxM5plate reader with SoftMax Pro 7.1 software and data is presented as 490 nm absorbance. For the LDH-Glo assay luminescent readings were taken on a SpectraMaxM5plate reader with SoftMax Pro 7.1 software, and data are presented as LDH concentration (µg/mL) calculated by standard curve.

**Single-cell RNA-sequencing sample preparation and sequencing.** In total, 30 asteroids per condition were pooled in a 1.5 mL Protein LoBind Eppendorf (Eppendorf cat#022-43-108-1) and allowed to settle. The supernatant was carefully discarded, and a single-cell suspension was produced by incubation in 500 µL digestion buffer

(Accutase™ with 80 U/mL Protector RNase Inhibitor (Sigma-Aldrich cat#03335402001) for 1 h at 37 °C with gentle pipette mixing every 10 min. At the end of the incubation the single-cell suspension was washed with 500 µL wash buffer (0.02% BSA in 1× PBS with 80 U/mL Protector RNase Inhibitor) and passed through a 20 µM filter (MACS, Miltenyi Biotec cat#130-101-812) to a fresh 2 mL Protein LoBind Eppendorf. An additional 1 mL of wash buffer was then passed through the same filter for a total single-cell suspension of 2 mL. The samples were centrifuged at 300×$g$ for 5 min at 4 °C followed by another wash in 1 mL wash buffer. After another centrifugation, the supernatant was discarded, and the single-cell pellet gently resuspended in 50 µL wash buffer. Cells were counted in quadruplicate on the Cellometer K2 with AOPI and processed through the single-cell RNA-sequencing pipeline from 10X Genomics, 3′ Version 3 (10X Genomic Chromium).

Briefly, the single-cell suspension was mixed with RT reaction mix to target an 8000-cell recovery and 75 µl was loaded onto a chromium microfluidics chip with 40 µL of barcoded beads and 280 µL of partitioning oil. The chip was run on the chromium controller, encapsulating a single cell and barcoded bead within individual oil droplets. Reverse transcription was performed within these individual oil droplets to produce barcoded cDNA. cDNA was then isolated by Silane DynaBeads (Thermo Fisher Scientific, DynaBeads MyONE Silane, cat# 37002D) before PCR amplification. Amplified cDNA cleanup and size selection was performed using SPRIselect beads (Beckman-Coulter, SPRIselect, cat# B23317) and cDNA quality was assessed by the High-Sensitivity DNA assay on the Agilent 2100 BioAnalyzer (Agilent, High-Sensitivity DNA Kit, cat# 5067-4626). Sequencing libraries were then prepared according to 10X specifications, including fragmentation, sequencing adaptor ligation, and sample index PCR. Between each of these steps, library cleanup and size selection were performed by SPRIselect beads. Final cDNA library quality was assessed by the Agilent BioAnalyzer High-Sensitivity DNA assay and the Qubit High-Sensitivity DNA assay and quality-confirmed libraries were sequenced on Illumina's NextSeq 500 or NOVAseq platform to a depth of 200 million paired-end reads. Single-cell RNA-sequencing data were produced from four independent experimental replicates.

### Data analysis

**Image analysis.** Images were captured by Carl Zeiss confocal LSM700 and confocal Zeiss LSM 880 with Airyscan. The immunofluorescence-stained DAPI-positive cells in each image of asteroids were quantified by ImageJ by automated cell counting. The staining intensity in immunofluorescence-labeled asteroids were measured by ImageJ. The intensity of MC1, TOMA2, CP13, AT8, ThioS, and Fluoro Jade B were normalized by the corresponding Tuj1 intensity. Schematics were created with BioRender.com.

**GraphPad Prism statistical analysis.** Statistical analyses and figures artwork were performed using GraphPad Prism version 9.00 for

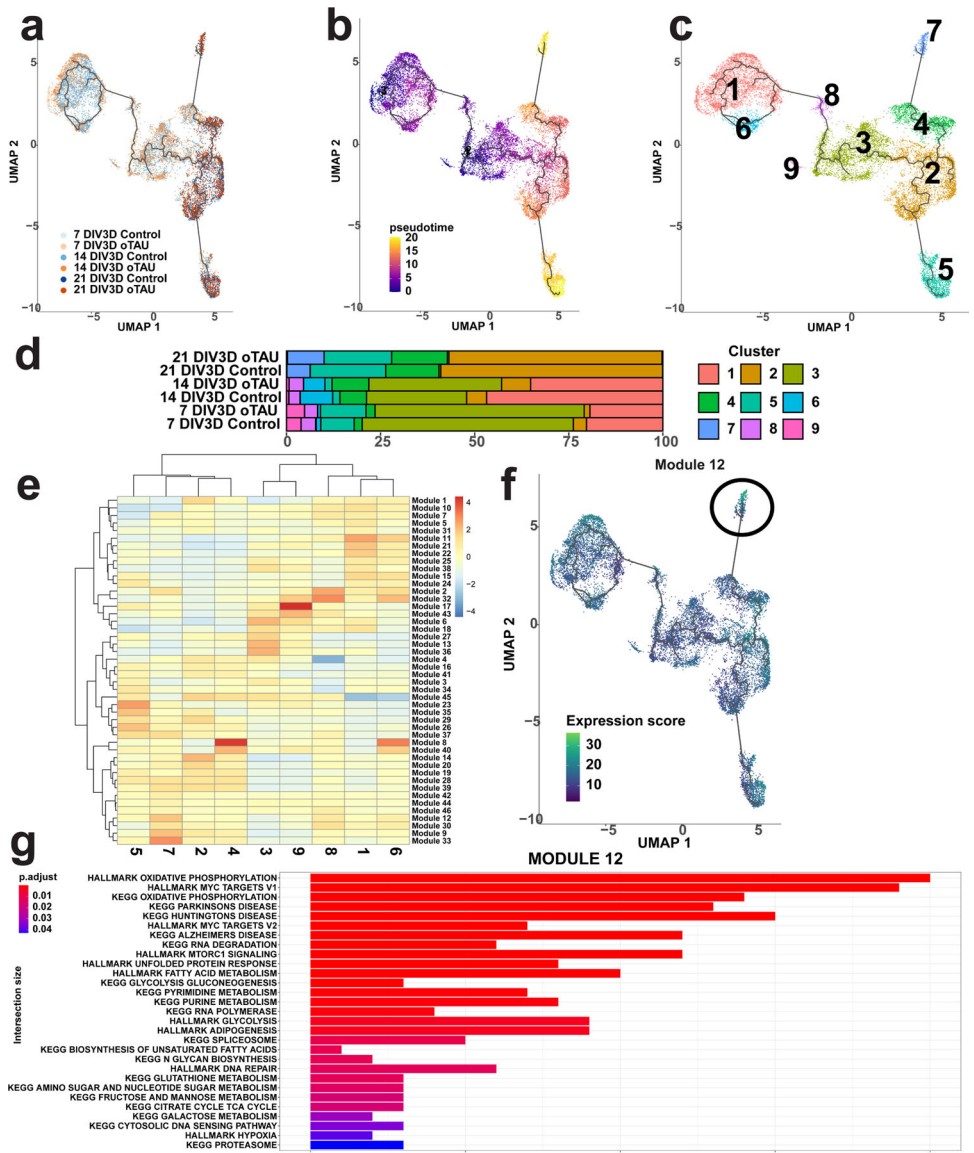

**Fig. 8 | Subpopulations of AstTau pathology-associated EX_NEU cells were identified by Monocle3 pseudotime trajectory and gene module signature analysis. a** UMAP of Monocle3 reclustering of only the EX_NEU cell population, colored by conditional timepoints across the 7–21 DIV3D control and AstTau time course (blue and orange gradients, respectively). **b** UMAP of the pseudotime trajectory of EX_NEU, representing the dynamic process of gene expression changes that occur during the progression of AstTau pathology. Note the two branches of late-stage yellow cell sub-clusters, indicative of a cell fate decision point. **c** UMAP of pseudotime clusters 1–9 produced by Monocle3 reclustering with pseudotime trajectory overlaid. **d** Percent composition of pseudotime clusters in control and AstTau across the 7–21 DIV3D time course revealing an overrepresentation of AstTau EX_NEU in cluster 7. **e** Modules of co-regulated differentially expressed genes within the pseudotime clusters, revealing module 12 upregulated within cluster 7. **f** UMAP feature expression of module 12 projected on the pseudotime trajectory. **g** Top 29 functional gene set enrichment pathways by adjusted *P* value of the differentially expressed genes defining module 12 (see "Methods"). Source data are provided as a Source Data file.

Windows with a two-sided α of 0.05. All group data are expressed as mean ± SEM. Column means were compared using one-way ANOVA with treatment as the independent variable. Group means were compared using two-way ANOVA with factors on oTau treatment and timepoints, respectively. When ANOVA showed a significant difference, pairwise comparisons between group means were examined by Tukey's, Dunnett or uncorrected Fisher's LSD multiple-comparison test. Significance was defined when *P* < 0.05. LDH assay data analysis was performed with a paired *t* test.

### Single-cell RNA-sequencing data analysis
**CellRanger pipeline.** CellRanger version 3.1.0 (10X Genomics) was used to combine and process the raw Illumina NextSeq 500 RNA and NOVAseq sequencing files. First each sequencing library was demultiplexed by sample index to generate FASTQ files for paired-end reads using the CellRanger mkfastq pipeline. FASTQ files were then passed to the CellRanger count pipeline, which used STAR aligner version 2.7[106] to align reads to the human reference genome (GRCh38). The CellRanger aggr pipeline was then used to equalize the aligned molecule_info.h5 sample libraries across sequencing depths (by each sample cell being down-sampled to have the same confidently mapped reads per cell) and aggregated together to generate the gene-cell barcode matrix. All subsequent data analysis was performed in R version 4.0.3.

**Seurat object filtration.** Subsequent filtering, normalization, and scaling of data was performed using Seurat version 4.1.1[107,108]. The Seurat object was created with a min.cells of 2 and a min.features of 200. Cells

with less than 200 and greater than 3000 detected genes or greater than 12% mitochondrial counts were filtered out to discard potential doublets and low-quality captures Gene counts for each cell were normalized by total expression, multiplied by a scale factor of 10,000 and transformed to log scale. PCA based on the highly variable genes detected (dispersion of 2) was performed for dimension reduction and the top 30 principal components (PCs) were selected based on a heuristic elbow plot and jack straw plot analysis. We clustered cells based on graph-based methods (KNN and Louvain community detection method) implemented in Seurat. Clusters were visualized using uniform manifold approximation and projection (UMAP)[109].

**Cluster cell-type identification.** To identify neuronal cell-type subpopulations, we performed differential expression analysis using the Wilcoxon rank-sum test implemented in Seurat between previously defined clusters with a min.pct of 0.25 and a logfc.threshold of 0.25. This identified top-expressing genes for each cluster, which were then considered alongside the feature expression of canonical gene cell-type markers to manually conclude cell-type cluster identification. In addition, the automated cell typing R package SingleR version 1.2.4 was used to confirm cell-type identification using integrated assay with the hpca.se$label.fine from the HumanPrimaryCellAtlasData[110].

**DE analysis.** Differential expression analysis was performed within each cell type between control and AstTau samples using the Wilcoxon rank-sum test implemented in Seurat with a min.pct of 0.1, a logfc.-threshold of 0.25, and a pseudocount of 1E4. A multiple-comparison correction was performed using the Benjamin & Hochberg FDR method to produce an adjusted $P$ value[111]. Differentially expressed genes were evaluated according to their log fold change (greater than log2(.25)) and adjusted $P$ values (<0.05). Figures were generated using the ggplot2 R package version 3.3.2[112].

**Functional enrichment analysis.** Functional gene set enrichment analysis of the significant differentially expressed genes between control and AstTau samples was performed using the R implemented GProfiler2 version 0.2.0[113]. The enrichment analysis was run as an ordered query (ordered by log2FC) using a α threshold of 0.05 and using Benjamin & Hochberg FDR for multiple testing correction[111]. Only genes in the Seurat dataset were considered by using a custom domain scope. A custom source GMT, gp_zSEF_sD9Q_d1M, was used. It includes all Hallmark gene sets, curated gene sets, and ontology gene sets from the Molecular Signatures Database (MsigDB) v7.2[70,114]. The enrichment analysis was assessed and visualized by a heatmap of significance (−log10($P$ value)) of the top 25 enriched pathways per sample comparison. Figures were generated using the ComplexHeatmap R package version 2.4.3[115].

Comparative functional gene set enrichment analysis between AstTau, AstαSYN and published datasets was performed with the same Grofiler2 settings, but as a non-ordered query. Gene input was as follows: From our AstTau dataset, common genes upregulated across IN_NEU, EX_NEU, and ASC at 21 DIV3D ($n = 11$, $P < 0.05$, log fold change >0.25). From our AstαSYN dataset, common genes upregulated across IN_NEU, EX_NEU, and ASC at 21 DIV3D ($n = 491$, p < 0.05, log fold change >0.25). From ref. 72, common genes upregulated across neurons and astrocytes by snRNA-seq of postmortem AD brain tissue (DEG 5 and 7, $n = 109$, log fold change >1 and FDR < 0.01). From ref. 74, common genes upregulated by snRNA-seq of postmortem AD brain tissue across AST, IN, and EX cell types in Fig. 2c, late versus early pathology ($n = 16$, log fold change >1 and FDR < 0.01)). From ref. 116, all highlighted genes upregulated by snRNA-seq of postmortem autism spectrum disorder (ASD) brain tissue across neuronal and non-neuronal cell types in Fig. 2a, b ($n = 22$, $P < 0.05$, log fold change >0.14). The enrichment analysis was assessed and visualized by a heatmap of significance (−log10($P$ value)) of the top ten unique

enriched pathways from AstTau, Grubman, and Mathys. All heatmaps were generated using the ComplexHeatmap R package version 2.4.3 and color scale generated using the dependent R package Circlize version 0.4.15[115]. Additional visualization of significant ($P < 0.05$) enriched pathways was performed using Cytoscape version 3.8.2 with EnrichmentMap version 3.3.4[117,118] with an edge cutoff of 0.375. Gene sets in EnrichmentMap cluster by similarity and annotations of shared gene set features were added manually using Cytoscape-implemented AutoAnnotate.

**Signature analysis.** The scaled expression per cell of literature curated and MsigDB ontology gene sets (see SourceFile) was compared between control and AstTau samples by computing the mean expression using colMeans and performing an ANOVA or t.test across comparison pairs using stat_compare_means. Figures were generated using the ggplot2 R package version 3.3.2[112].

**Liger integration.** To compare cell types of our data with a different dataset[38] we merged the two datasets using Liger version 1.0.0[119], which uses integrative Non-Negative Matrix Factorization (iNMF) to simultaneously reduce data dimensionality as well as correct the technical factors. We used Seurat v4.0.4[120] as described in "Methods" Seurat Object Filtration for pre-processing before running Liger. Each dataset is normalized to account for differences in total gene-level counts across cells using the NormalizeData function. Furthermore, we used the ScaleData function with parameter do.center = FALSE, to scale normalized datasets without centering by the mean, giving the nonnegative input data required by iNMF. To run Liger, we tuned the two most important parameter k (determines the number of matrix factors in the factorized data) and lambda (the degree of dataset integration) using the suggestK and the suggestLambda function, respectively. Finally, we used k = 20 and lambda = 5 to integrate the datasets. The resulting integrated dataset was then analyzed using UMAP visualizations and percent cluster cell count calculations as described in "Cluster cell-type identification".

**Monocle3 pseudotime trajectory analysis.** To identify AstTau-associated cell subtypes we isolated cell-type clusters and reprocessed using the Monocle3 package version 1.0.0[79–83]. Briefly, the cell-type-specific subset Seurat object was converted to a Monocle object and processed by the standard dimensional reduction and cell clustering function with a resolution of 1e-4. The pseudotime trajectory was inputted by the learn_graph call with use_partition = F. Cells were then ordered in pseudotime and differential gene expression testing was performed to identify cluster-specific markers and gene modules using the find_gene_modules function. Visualizations were produced using built-in Monocle3 functions. Functional gene set enrichment was performed as described in "Methods" Functional enrichment analysis using the GProfiler2, with the only change being the query was limited to HALLMARK and KEGG MsigDB pathways. Results were visualized using the enrichplot version 1.10.2 and Dose version 3.16.0 packages[121,122].

## Statistics and reproducibility

No statistical method was used to predetermine the sample size. Sample size determination for immunolabeling quantification was based on prior studies with 3D assembloid modeling[28,32,37]. High-throughput scRNA-seq was selected to profile >1000 cells per replicate experimental condition in order to provide a sufficiently large sample size for analysis.

No data were excluded from the immunolabeling analyses. Low-quality and multiplet cell reads were removed using standard quality control filtration in the analysis of the scRNA-seq data, excluding cells with less than 200 and greater than 3000 detected genes, or greater than 12% mitochondrial counts.

All immunolabeling experiments were repeated in at least three independent batches of asteroid cultures with at least five individual asteroids per quantification. scRNA-seq was repeated in four independent batches of asteroid cultures. Cultured 3D asteroids within a batch were blindly and randomly selected at timepoint collection for immunolabeling or scRNA-seq. In addition, well plates were randomized for treatment to avoid marginal effects of cell growth on the plate. Further covariate controls are not relevant to this study due to the highly controlled nature of the culture system. Quantification of transcriptomics, granular intensity, MAP2 dendritic length, and immunoblot band intensity were blindly repeated by co-authors.

### Reporting summary
Further information on research design is available in the Nature Research Reporting Summary linked to this article.

## Data availability
Raw and processed scRNA-seq data generated in this study have been deposited in the NCBI Gene Expression Omnibus database (GEO) under accession code GSE186356. Processed scRNA-seq datasets generated in this study have been deposited on the Single Cell Portal, including the cell barcodes, UMAP coordinates, and other available characteristics [https://singlecell.broadinstitute.org/single_cell/study/SCP1621/asteroid1-2021]. Publicly available datasets used in this study are available at The Molecular Signatures Database [https://www.gsea-msigdb.org/gsea/msigdb] and The Human Primary Cell Atlas [http://biogps.org/dataset/2429/primary-cell-atlas/]. Source data are provided in the Source Data files as follows: Source data of differential gene expression results for Figs. 4c and 7b, cell-type markers for Supplementary Fig. 5c, and uncropped immunoblots for Fig. 3c, GProfiler2 FGSA results and gene set names for Figs. 5a, 7e, 8g, Supplementary Figs. 14h, 15, and 16, GProfiler2 FGSEA results in Cytoscape formatting for Fig. 5b and Supplementary Fig. 17a, b, gene set names and GProfiler2 FGSEA results for Fig. 5c, and module scores for Fig. 8e and Supplementary Fig. 14e are available in Supplementary File SourceFile. Detailed explanations of gene sets used Figs. 4d, e and 7c, d are also available in Supplementary File SourceFile. Source data are provided with this paper.

## Code availability
The original R scripts for Seurat processing are available on GitHub [https://github.com/satijalab/Seurat]. All custom code to reproduce the analyses and figures reported in this paper is available on GitHub [https://doi.org/10.5281/zenodo.7102480][123].

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

## Acknowledgements

We thank Dr. Todd A. Blute for his technical support at BU's Proteomics and Imaging Core Facility. We thank Rakez Kayed (UT Galveston) for the provision of the TOMA2 antibody, Nicholas M. Kanaan (Michigan State University) for the provision of the Tau13 antibody (originally created by Lester I Binder, Northwestern University), and Peter Davies (Northwell/Hofstra, deceased) for provision of CP13 antibody. We would like to thank the following funding agencies for their support: B.W. & C.S.C: Kilachand Fund of the BU Center for Life Sciences Engineering, B.W.: NIH (AG050471, AG056318, AG064932, AG061706, AG072577, AG071200) and the BrightFocus Foundation, C.S.C: NIH (DA047032, DA051411, DA051889, DA050243, AG074591, DA056006, DA047032), M.M: NS125469-01A1.

## Author contributions

H.D.R. designed and performed the experiments and bioinformatics analysis, interpreted the results, and wrote the paper. L.J. designed, performed, and analyzed the experiments, interpreted the results, and wrote the paper. R.H. helped design statistical tests for scRNA-seq analysis. N.K.O.'N. helped design statistical tests for scRNA-seq analysis. M.M., C.A.M., and B.J.S. helped perform experiments. L.Z. helped analyze the experiments. D.S. helped design scRNA-seq analysis. B.W. conceived and oversaw the project, provided guidance, interpreted the results, and wrote the paper. C.S.C. conceived and oversaw the project, provided guidance, interpreted the results, and wrote the paper.

## Competing interests

The authors declare the following competing interests: B.W. is co-founder and Chief Scientific Officer for Aquinnah Pharmaceuticals Inc. B.W., C.S.C., L.J., and H.D.R. are listed as joint inventors on a provisional patent filing regarding oligomer seeding in the iPSC-derived assembloid model: "Synthetic 3D Brain Organoids and Uses Thereof"; US Provisional Application No.: 63/314,585, Inventor(s) Wolozin, Cheng, Jiang, and Rickner with patent applicant The Trustees of Boston University. The remaining authors declare no competing interests.
