## [Peer Review File · Nature Communications]

Single cell transcriptomic profiling of a neuron-astrocyte assembloid tauopathy modelREVIEWER COMMENTS

Reviewer #1 (Remarks to the Author):

Rickner and Jiang et al. developed a 3D model relevant to neurodegeneration observed in AD. The model is based on a previously published study that the authors have cited where neurons and astrocytes were differentiated from human iPSC, and then co-cultured in 3D. The authors implemented this model by differentiating neurons using NGN2 induction in neural progenitors. Astrocytes were differentiated using small molecules. Neurons in 2D were treated with mouse lysates enriched in oligomerized Tau, then co-cultured with astrocytes to form spheroids, termed asteroids. The authors observed progressive increase in the Tau pathology namely, hyperphosphorylation, misfolding and fibril formation in the asteroids, accompanied by increase in neuronal degeneration and astrogliosis. They performed single cell RNA-seq analysis of the treated and non-treated asteroid cultures at day 7, 14 and 21 after inducing the 3D cultures. Single cell analysis revealed activation of inflammatory pathways, ribosomal proteins and changes in a few heat shock proteins with differential activation of these pathways between excitatory and inhibitory neurons. Treatment with the HSP90 inhibitor reduced neuronal degeneration and Tau pathology. Additionally, HSP90 inhibition also reverted transcriptional changes in the neurons.

The paper has several interesting advances including a 3D neuron-astrocyte culture relevant to AD pathology as well as single cell transcriptomic analysis of the asteroids over a time course of degeneration. However, though the premise of the study is very interesting, the phenotypic analysis of the asteroids and the subsequent single cell analysis could have been more detailed. From the phenotypic analysis, it wasn't clear why a 3D culture is required to generate a cell death phenotype. These phenotypes have been observed using 2D cultures that are far more reproducible than 3D cultures. The single cell analysis also did not reveal anything significantly more than what was already known. The role of chaperone proteins, especially the heat shock proteins is well known in propagating Tau pathology. HSP90 inhibition has been found to reduce Tau pathology almost ten years ago (PMID: 25069659).

Given their model and the single cell analysis, I would have expected either more on the phenotypic side or a detailed analysis on the single cell transcriptomic side. I have listed my concerns below:

1. The authors claim that their 3D system is "mature". The neuronal morphology shown in the Fig S1 looks very immature. Have the authors performed any functional tests to ascertain that these neurons are electrically active? Also, are the astrocytes functionally active? Supplementary video 1 shows a single neuron with minimal branching and it is difficult to see the arborisation and interactions claimed by the authors. Neuron-astrocyte co-cultures have been performed previously in 2D. Based on the video provided, it cannot be claimed that the 3D cultures results in a more mature phenotype.
2. One key issue with 3D cultures is the inconsistency across spheroids. The authors need to provide data that shows how variable the sphere composition is within and across differentiations.
3. In Fig. 1f, the inhibitory neuronal population seems stable in the 3D cultures but the excitatory neuronal populations keeps increasing from as low as ~10% at 7 DIV3D to ~40% by 21 DIV3D. This is unexpected because NGN2 induces mainly excitatory neurons from NPCs. So I would have expected a larger percentage of excitatory neurons compared to inhibitory neurons in their starting cultures. This also means that a significant proportion of cells at the phenotypic time point are immature as they are newly born. It is not clear whether the phenotypes observed are due to the immature or mature population. The authors measure maturity but it is unclear how this was measured.
4. In Fig. 1j, the clustering has changed and does not correspond to Fig. 1c. hence, it is not possible to assess which cells have been deemed as mature.

5. Pg 6. As far as I now, TOMA antibodies recognize oligomeric tau while MC1 recognizes conformationally changed Tau.

6. Fig 3i: The MC1 signal needs to be co-localized with the GFP signal and then quantified to support the conclusion.

Phenotypic analysis needs to be expanded. An early hallmark of AD is the loss of synapses. Can the authors recapitulate this phenotype in their 3D culture? As I have mentioned above, it is not clear why a 3D culture is required in the first place. Are the authors suggesting that the observed phenotypes cannot be seen in 2D cultures? Are astrocytes even required to propagate the phenotype? The S100b staining would suggest that gliosis occurs earlier than the Fluro Jade staining but is that a technical artefact of detection limits or is astrocytic pathology required or does it accelerate neuronal dysfunction? Do astrocytes also show oTau pathology? And at what stage?

7. Fig. 4: The authors claim that "all datasets are integrated". This needs to be evidenced by showing the consistency of cell type proportions across replicates.

8. Translational stress response is usually accompanied by an inhibition of the ribosome biogenesis. Here ribosomal transcripts seem to be upregulated. It is not clear why the authors claim this a stress response.

9. The authors should refer to Fig S11 to make the claim that neuroinflammation is enriched in EX_NEU as Fig 4C does not show this.

10. Does A1, A2 and AD-associated astrocytic signatures arise only by 21 DIV3D or does the signature arise earlier?

11. The p-values are deemed significant even though the observed differences are very small. Using this approach will assign a large number of pathways as being differentially activated in each sample type. Why did the authors choose to focus on these 3-4 pathways shown?

12. Do they observe enrichment of familial AD associated genes in the upregulated or downregulated gene sets?

13. Since they have the single cell profiles and a measure of maturity, do they see a correlation of the observed phenotypes with the maturity of the neurons? Can they order the neurons along a pseudotime of degeneration either by using all the differentially expressed genes or genes involved in apoptosis?

14. Fig 7f does not add much to the analysis without a comparison across EX_NEU and IN_NEU. I would recommend either removing this panel or adding the corresponding analysis for both EX_NEU and IN_NEU across the different time points.

15. In the discussion, the authors claim their's the first model to show a neurodegeneration phenotype in response to oTau. This is not true. Please see PMID: 26490863. Also, can they highlight the figure that shows a progressive loss of glutamatergic neurons?

Reviewer #2 (Remarks to the Author):

This manuscript by Rickner et al. describes the generation of a novel 3D neuron-astrocyte co-culture system for investigating tauopathies (AstTau). The authors demonstrate that these cultures, when seeded with mouse brain homogenate over-expressing 4R1N P301S human Tau, are capable of developing pathology seen in

tauopathies. These pathologies include the presence of tau fibrils and oligomers as well as neurodegeneration. In addition, the authors characterized the effects of tau on the astrocytes through scRNAseq and showed that these astrocytes have some transcriptomic features of disease-relevant states of astrocytes. Through single cell transcriptomics, the authors found an upregulation of genes in the heat shock protein chaperone system in AstTau compared to unseeded controls. They were then able to mitigate the phenotypes associated with tauopathies using a heat shock protein inhibitor, PU-H71.

This study shows a novel and easily manipulable method that has the potential to model spreading of neurodegenerative disease pathology in human derived cells in a manageable time frame. In addition, the authors were able to identify a putative molecular candidate that reversed many facets of neurodegenerative disease including the level of oligomeric Tau and toxic astrocytic signatures. Although this model is useful and a step towards modeling neurodegenerative diseases, several concerns should be addressed to make this manuscript suitable for publication.

Major concerns:

There needs to be further validation and characterization of the AstTau model. The authors show through single cell sequencing that there is a large population of inhibitory neurons generated from NGN2 induction (Figure 1). This in itself seems quite novel as NGN2 induction is commonly associated with generating excitatory populations (Lin et al., 2021, Stem Cell Reports; Zhang et al., 2013, Neuron) and therefore would benefit from orthogonal methods to validate its presence, such as with staining AstTau with an inhibitory neuron marker (e.g. GAD65/67) and an excitatory neuron marker (e.g. VGLUT1).

There are several images where there seems to be significant overlap between the TUJ1 and S100B staining (e.g. Supplemental Fig 1D). It may be beneficial to explore whether this is a real biological phenomenon or an artifact of staining, especially for image quantification purposes.

Some considerations need to be made in terms of image segmentation and quantification. Authors show MC-1, TOMA2A, AT8, ThioS signal intensity as normalized by number of DAPI+ cells which stains all cells. It would be beneficial to be able to segment based on whether MC1, TOMA2A, ThioS is inside of neurons or astrocytes inside of this co-culture system or trapped somewhere in the extracellular space, particularly as astrocytes could also be spreading tau, perhaps using a similar method as described in figure 3K, or by labeling astrocytes and neurons with different fluorescent proteins. In addition, although the total number of cells are accounted for, the proportion of glia to neurons may be different from AstTau to AstTau and it would be interesting to assess whether that contributes to differences in Tau spreading.

In several images, antibody stains seem to be concentrated at the periphery of spheroids. This raises the question whether antibody penetration into the spheroid contributes to the observed patterns. The authors should validate some of the key patterns by staining sections of the spheroids (as opposed to wholemounts)

The authors invoke the epichaperome model, which has mostly been proposed by a single lab. The data shown in Fig. 6a is not convincing - Hsc70 (which is constitutively expressed) is undetectable as a 70 kDa band in untreated AstTau, which raises questions as to what the antibody detects. If the authors want to make claims about the epichaperome in their model, additional, orthogonal biochemical characterization is needed to support this claim. A simpler interpretation may be that Hsp90 inhibitors promote more Hsf1-mediated chaperone production (as seen in Fig. 7), which is generally thought to be beneficial. The authors could use different Hsp90 inhibitors to test this hypothesis.

Minor Concerns:

- In Fig. 7b, clarify whether the log₂FC in transcript levels refers to a change between oTau treated vs. untreated samples
- Labels in several figures are extremely small. Please reformat figures to make them more legible.
- Please correct "NOVA-seq" in methods to NovaSeq.

RESPONSES TO REVIEWER COMMENTS

To the reviewers:

We appreciated the reviewers' positive comments about the manuscript, including noting important "advances in 3D-neuron-astrocyte culture relevant to AD", and "a novel and easily manipulable method that has the potential to model spreading of neurodegenerative disease pathology in human derived cells in a manageable time frame."

We are pleased to report that we have now addressed all of the concerns raised by the reviewers and present a description of our responses below.

Reviewer #1:

1. The single cell analysis also did not reveal anything significantly more than what was already known. The role of chaperone proteins, especially the heat shock proteins is well known in propagating Tau pathology. HSP90 inhibition has been found to reduce Tau pathology almost ten years ago (PMID: 25069659).
 - You are absolutely correct, heat shock chaperones proteins have been well established to play a role in tau mediated neurodegeneration, even recently highlighted by single cell RNA sequencing of post-mortem AD brain tissue, so we were excited to see the heat shock response in our system as it demonstrated that our model is accurately recapitulating the processes associated with neurodegeneration as our goal is to produce a model that recapitulates AD and is responsive to neuroprotective treatments. We also demonstrated neuroinflammatory states that differed between excitatory and inhibitory neurons and in subsequent analysis have identified glutamatergic neuron and astrocyte subpopulation signatures that are associated with neurodegeneration (Suppl. Fig. 14).
2. The authors claim that their 3D system is "mature". The neuronal morphology shown in the Fig S1 looks very immature. Have the authors performed any functional tests to ascertain that these neurons are electrically active?
 - The reviewer is indeed correct, and we apologize for lack of clarity in the figure legend. The cells in Fig S1 are the iPSC derived neuronal precursors before integration into 3D culture, where integration and continued maturation occurs. Thus, they are indeed expected to look less mature. However, once in 3D culture the cells continue to mature. We show the maturation through expression of markers for mature neurons, such as MAP2, NeuN, PSD95, SYP, VGLUT1, and GAD67 (Fig. 1b, Suppl. Fig. 1).
 - In addition, we greatly appreciate the reviewer's suggestion that we perform functional tests to determine electrical activity. We did this using whole-cell patch clamp electrophysiological recording and intracellular filling of neurons and are pleased to report evidence of spontaneous synaptic events and of neuronal processes and synaptic structures in AstTau. These data are now presented as Suppl. Fig. 4.
3. Also, are the astrocytes functionally active?
 - Our data show strong responses of the astrocytes in the AstTau model. These changes include expression of reactive astrocytic marker GFAP (Fig. 1b). The

astrocytes also exhibit strong astrocytosis and extension of processes in AstTau, which further indicates functional activity (Suppl. Fig. 6).

4. Supplementary video 1 shows a single neuron with minimal branching and it is difficult to see the arborization and interactions claimed by the authors. Neuron-astrocyte co-cultures have been performed previously in 2D based on the video provided, it cannot be claimed that the 3D cultures results in a more mature phenotype.
 - Thank you for the feedback on the videos, we have improved this analysis with high resolution imaging of 10 μm sections. This high-resolution imaging shows the intimate proximity of the neurons and astrocytes, which complements the extensive arborization and interactions shown in Suppl. Fig. 2b, that develop in the AstTau model as compared to 2D culture (Suppl. Fig. 2).
 - We also show a comparison of arborization in 2D vs. 3D cultures (Suppl. Fig. 2a) that demonstrates the strong increase in arborization upon growth in 3D.
5. One key issue with 3D cultures is the inconsistency across spheroids. The authors need to provide data that shows how variable the sphere composition is within and across differentiations.
 - You are correct that classic organoids exhibit inconsistent growth. However, our asteroids exhibit structures and morphologies that are remarkably consistent among different experiments. We now explicitly demonstrate this in the manuscript with figures and graphs, providing images of replicate asteroid across batches, as well as an analysis of cell type composition across batches (Suppl. Fig. 6).
 - Most commonly used 3D culture models of cerebral organoids which are induced directly from the iPSC stage exhibit culture to culture variability and inconsistency. In contrast, the asteroids that we use are neuron/astrocyte assembloids that allow strong control over cell type differentiation and model composition. Thus, the AstTau model approach provides advanced control that enables more reproducible cultures and results across different experiments and cultures. This is an important advance presented by our manuscript.
6. In Fig. 1f, the inhibitory neuronal population seems stable in the 3D cultures but the excitatory neuronal populations keep increasing from as low as ~10% at 7 DIV3D to ~40% by 21 DIV3D. This is unexpected because NGN2 induces mainly excitatory neurons from NPCs. So I would have expected a larger percentage of excitatory neurons compared to inhibitory neurons in their starting cultures. This also means that a significant proportion of cells at the phenotypic time point are immature as they are newly born. It is not clear whether the phenotypes observed are due to the immature or mature population. The authors measure maturity but it is unclear how this was measured.
 - The reviewer has astutely noted classic points stemming from the nature of iPSC culture modeling. One of the strengths of the AstTau system is that it does indeed generate multiple neuronal cell types (including inhibitory neurons). While the precise mechanism responsible for this technological advance remains to be determined, it seems possible that the increased phenotypic diversity results from the use of partial NGN2 induction and the subsequent integration with astrocytes in 3D culture. We also note that NGN2 overexpression has recently been demonstrated to produce a broader range of neuronal phenotypes than originally thought¹, which we are able to more finely dissect with scRNA-seq. Our data

further show that the immature precursor population that is initially present in AstTau undergo further maturation, with this immature population dramatically shrinking over the 21 DIV3D time course, resulting in more mature neurons that are able to recapitulate neurodegeneration (Fig. 1, Suppl. Fig 5).

7. In Fig. 1j, the clustering has changed and does not correspond to Fig. 1c. hence, it is not possible to assess which cells have been deemed as mature.
 - We apologize for this confusion. The clustering in Fig. 1j is a different integration than the clustering in Fig 1c. To perform a comparison of maturity, our data (from the clustering in Fig. 1c). was reintegrated with datasets from other 3D cerebral organoid models, and clustering, cell typing, and cluster composition analysis was performed on the new integration to identify mature cell clusters and what dataset was overrepresented in them. This analysis has been expanded on and the following text has been added to the results on page 5 to provide clarity: “Clustering, cell type identification, and cluster composition analysis was performed on the new integrated dataset and revealed that the asteroid model (red) was overrepresented in more mature cell type clusters compared to the cerebral organoid model (blue)...” (Suppl. Fig. 7).
8. Pg 6. As far as I know, TOMA antibodies recognize oligomeric tau while MC1 recognizes conformationally changed Tau.
 - The reviewer is correct! We have changed the language on page 6 of the manuscript to accurately reflect this distinction: “Misfolding and oligomerization of tau were observed using the MC1 and TOMA2 antibodies respectively...”.
9. Fig 3i: The MC1 signal needs to be co-localized with the GFP signal and then quantified to support the conclusion.
 - We thank the reviewer for an excellent suggestion. We have performed this analysis and added language on page 7 of the manuscript to describe the results: “Colocalization of the NGN2-GFP and MC1 signal returned a Pearson’s R coefficient of correlation of 0.09 as compared to a 0.89 correlation of Tuj1 and MC1 signal in the same spheroids, further validating that a subset of unexposed GFP+ neurons were subject to oTau propagation (Fig. 3n).:”
10. Phenotypic analysis needs to be expanded. An early hallmark of AD is the loss of synapses. Can the authors recapitulate this phenotype in their 3D culture?
 - This is an important additional to the phenotypic analysis, thank you for the suggestion. We have now performed VGLUT1 immunolabeling and quantified glutamatergic synaptic loss. The resulting data provide additional information supporting the utility of our system in recapitulation neurodegeneration (Suppl. Fig. 10 d-e).
11. As I have mentioned above, it is not clear why a 3D culture is required in the first place. Are the authors suggesting that the observed phenotypes cannot be seen in 2D cultures?
 - The reviewer’s comment suggests that the manuscript would benefit from language that clearly states the benefits of the 3D system. To this end, we have added figures comparing the neuron/astrocyte cultures in 2D versus 3D (showing the increased complexity of arborization (Suppl. Fig. 2) and added the following text on page 3: “The resulting neurons and astrocyte disease model rapidly recapitulates physiologically relevant spatial cellular interactions...”

- Numerous papers and reviews have identified the benefits of using 3D culture systems²⁻⁴. In our hands, we have demonstrated that the neurons and astrocytes show much closer interactions in 3D than in 2D culture (Suppl. Fig. 2). Additionally, the 3D cultures system evolves neurodegeneration over weeks rather than hours, which enables robust and detailed transcriptional and pathophysiological studies as well as allowing time for the astrocytes to respond to the neurodegeneration, for instance with astrogliosis.
12. Are astrocytes even required to propagate the phenotype? The S100b staining would suggest that gliosis occurs earlier than the Fluro Jade staining but is that a technical artefact of detection limits or is astrocytic pathology required or does it accelerate neuronal dysfunction? Do astrocytes also show oTau pathology? And at what stage?
 - Astrocytes are necessary to the production and continuity of the 3D AstTau model (Suppl. Fig. 2). Indeed, it is not possible to study the system without astrocytes because the neurons do not maintain a viable 3D structure. We do note that astrocyte pathology precedes neurodegeneration by morphological changes and S100 β staining, adding to the growing body of evidence that astrocytes play a critical role in the propagation of neurodegeneration phenotypes. The AstTau model is a powerful platform that enables dissection of astrocytic contributions, which we demonstrate in this manuscript.
 13. Fig. 4: The authors claim that “all datasets are integrated”. This needs to be evidenced by showing the consistency of cell type proportions across replicates.
 - The reviewer makes an excellent point. We have now added an analysis of cell type composition across batches by immunolabeling and scRNA-seq (Suppl. Fig. 6) to support this.
 14. Translational stress response is usually accompanied by an inhibition of the ribosome biogenesis. Here ribosomal transcripts seem to be upregulated. It is not clear why the authors claim this a stress response.
 - This is an interesting point. The reviewer is correct that ribosomal biogenesis is inhibited, but this generally refers to ribosomal proteins. To address the question of a translation stress response, we have performed immunolabeling of stress granule marker HNRNPA2B1. Stress granules formation is associated with the translational stress response and translational inhibition. We see an increase in stress granule immunolabeling at the 7 DIV3D timepoint at which ribosomal transcripts are upregulated (Suppl. Fig. 10g-h). We hypothesize that the reduction in protein synthesis is not reflected by a corresponding reduction in ribosomal transcripts.
 15. The authors should refer to Fig S11 to make the claim that neuroinflammation is enriched in EX_NEU as Fig 4C does not show this.
 - Thank you for catching this! Changes have been made to reflect the proper reference in Fig. 4 and Suppl. Fig 16.
 16. Does A1, A2 and AD-associated astrocytic signatures arise only by 21 DIV3D or does the signature arise earlier?
 - The A1, A2, and AD-associated astrocytic signatures are significant at 21 DIV3D only.
 17. The p-values are deemed significant even though the observed differences are very small. Using this approach will assign a large number of pathways as being

differentially activated in each sample type. Why did the authors choose to focus on these 3-4 pathways shown?

- You are correct that GSEA produces a large number of pathways, and many are overlapping as determined by Cytoscape GSEA enrichment map analysis (Fig. 5, Suppl. Fig. 17). The pathways chosen for focused analysis and discussion are representative of the most highly represented groups of these overlapping pathways from the Cytoscape GSEA enrichment map analysis (Fig. 5, Suppl. Fig. 17).
18. Do they observe enrichment of familial AD associated genes in the upregulated or downregulated gene sets?
- Really interesting question! Since the AstTau model is focused on the contribution of oTau to the development of pathology in non-familial AD iPSCs we had not performed that analysis. At your suggestion we have mapped expression changes of familial AD associated GWAS genes between conditions in AstTau at 21 DIV3D (Fig. 7d).
19. Since they have the single cell profiles and a measure of maturity, do they see a correlation of the observed phenotypes with the maturity of the neurons? Can they order the neurons along a pseudotime of degeneration either by using all the differentially expressed genes or genes involved in apoptosis?
- Thank you for this suggestion, this is a fantastic way to broaden our analysis of neurodegeneration in our dataset we had not tried. We have added this analysis and discussion in the manuscript (Fig 8).
20. Fig 7f does not add much to the analysis without a comparison across EX_NEU and IN_NEU. I would recommend either removing this panel or adding the corresponding analysis for both EX_NEU and IN_NEU across the different time points.
- This is a great point; the comparison is powerful. Due to space constraints, this comparison has been added as a supplemental figure (Suppl. Fig 17).
21. In the discussion, the authors claim theirs's the first model to show a neurodegeneration phenotype in response to oTau. This is not true. Please see PMID: 26490863.
- Thank you for identifying this misstatement, we mean to highlight the advance our model makes in demonstrating a neurodegenerative phenotype in response to oTau in a 3D neuron-astrocyte co-culture system. We have corrected the language in the manuscript and ensured we have incorporated your useful references.
22. Also, can they highlight the figure that shows a progressive loss of glutamatergic neurons?
- We have added a figure to highlight this loss and associated pathways! (Fig 8, Suppl. Fig. 10d-f).

Reviewer #2:

Major Concerns:

1. There needs to be further validation and characterization of the AstTau model. The authors show through single cell sequencing that there is a large population of inhibitory neurons generated from NGN2 induction (Figure 1). This in itself seems quite novel as NGN2 induction is commonly associated with generating excitatory populations (Lin et al., 2021, Stem Cell Reports; Zhang et al., 2013, Neuron) and therefore would benefit

from orthogonal methods to validate its presence, such as with staining AstTau with an inhibitory neuron marker (e.g. GAD65/67) and an excitatory neuron marker (e.g. VGLUT1).

- As discussed with reviewer 1, we use a NGN2 transduction protocol that produces neuronal cultures with selective NGN2 expression allowing for the development of excitatory and inhibitory populations. Additionally, NGN2 overexpression has recently been demonstrated to produce a broader range of neuronal phenotypes than originally thought¹, which we are able to more finely dissect with scRNA-seq. Based on your suggestion we have performed immunolabeling of inhibitory and excitatory neuronal markers to validate these populations (Suppl. Fig. 1d-e).
2. There are several images where there seems to be significant overlap between the TUJ1 and S100B staining (e.g. Supplemental Fig 1D). It may be beneficial to explore whether this is a real biological phenomenon or an artifact of staining, especially for image quantification purposes.
 - You are absolutely correct, this overlap is indicative of the close interactions between the neurons and astrocytes. We have further demonstrated these interactions, and individual localization of cell types demonstrating it is not an artifact of staining, by high resolution imaging (Suppl. Fig 2c).
 3. Some considerations need to be made in terms of image segmentation and quantification. Authors show MC-1, TOMA2A, AT8, ThioS signal intensity as normalized by number of DAPI+ cells which stains all cells. It would be beneficial to be able to segment based on whether MC1, TOMA2A, ThioS is inside of neurons or astrocytes inside of this co-culture system or trapped somewhere in the extracellular space, particularly as astrocytes could also be spreading tau, perhaps using a similar method as described in figure 3K, or by labeling astrocytes and neurons with different fluorescent proteins. In addition, although the total number of cells are accounted for, the proportion of glia to neurons may be different from AstTau to AstTau and it would be interesting to assess whether that contributes to differences in Tau spreading.
 - Thank you for these suggestions, we have revised quantification to address cell type specific staining (Fig 2, 3, 6).
 4. In several images, antibody stains seem to be concentrated at the periphery of spheroids. This raises the question whether antibody penetration into the spheroid contributes to the observed patterns. The authors should validate some of the key makers by staining sections of the spheroids (as opposed to whole mounts).
 - We apologize for presenting figures that gave rise to the impression of incomplete penetration. We have now replaced panels in figure 2 that clearly give antibody labeling throughout the section. In addition, we performed immunolabeling of thin sections (10 μ m, Suppl. Fig. 9f), and observed similar labeling patterns as the images shown in figure 2.
 5. The authors invoke the epichaperome model, which has mostly been proposed by a single lab. The data shown in Fig. 6a is not convincing - Hsc70 (which is constitutively expressed) is undetectable as a 70 kDA band in untreated AstTau, which raises questions as to what the antibody detects. If the authors want to make claims about the epichaperome in their model, additional, orthogonal biochemical characterization is needed to support this claim. A simpler interpretation may be that Hsp90 inhibitors promote more Hsf1-mediated chaperone production (as seen in Fig. 7), which is generally

thought to be beneficial. The authors could use different Hsp90 inhibitors to test this hypothesis.

- Great points, thank you for the alternative interpretation. Based on continuing exploration of the epichaperome in the AstTau model we agree that there are alternative explanations for the role of heat shock proteins and heat shock protein inhibition in the AstTau system. The PU-H71 data emphasize the importance of the HSF1 driven HSP chaperone involvement in tau pathology, but we decided to remove any discussion of a putative “epichaperome”.

Minor Concerns:

1. In Fig. 7b, clarify whether the log2FC in transcript levels refers to a change between oTau treated vs. untreated samples.
 - Apologies for the confusion, you are correct the log2FC refers to the change between oTau and untreated samples. Clarification has been added to the figure legend.
2. Labels in several figures are extremely small. Please reformat figures to make them more legible.
 - Apologies, figures have been reformatted to address this issue.
3. Please correct “NOVA-seq” in methods to NovaSeq.
 - Good catch and thank you again for all your constructive comments!

References

1. Lin, H.C. *et al.* NGN2 induces diverse neuron types from human pluripotency. *Stem Cell Reports* **16**, 2118-2127 (2021).
2. Choi, S.H., Kim, Y.H., Quinti, L., Tanzi, R.E. & Kim, D.Y. 3D culture models of Alzheimer's disease: a road map to a “cure-in-a-dish”. *Molecular neurodegeneration* **11**, 75-75 (2016).
3. Venkataraman, L., Fair, S.R., McElroy, C.A., Hester, M.E. & Fu, H. Modeling neurodegenerative diseases with cerebral organoids and other three-dimensional culture systems: focus on Alzheimer’s disease. *Stem Cell Reviews and Reports* (2020).
4. Hedegaard, A., Stodolak, S., James, W.S. & Cowley, S.A. Honing the Double-Edged Sword: Improving Human iPSC-Microglia Models. *Frontiers in Immunology* **11**(2020).

REVIEWER COMMENTS

Reviewer #1 (Remarks to the Author):

The authors have addressed most of my concerns either through experiments or clarifications. There are a few minor changes regarding some claims I would like to see before acceptance.

1. The single cell analysis also did not reveal anything significantly more than what was already known. The role of chaperone proteins, especially the heat shock proteins is well known in propagating Tau pathology. HSP90 inhibition has been found to reduce Tau pathology almost ten years ago (PMID: 25069659).

• You are absolutely correct, heat shock chaperones proteins have been well established to play a role in tau mediated neurodegeneration, even recently highlighted by single cell RNA sequencing of post-mortem AD brain tissue, so we were excited to see the heat shock response in our system as it demonstrated that our model is accurately recapitulating the processes associated with neurodegeneration as our goal is to produce a model that recapitulates AD and is responsive to neuroprotective treatments. We also demonstrated neuroinflammatory states that differed between excitatory and inhibitory neurons and in subsequent analysis have identified glutamatergic neuron and astrocyte subpopulation signatures that are associated with neurodegeneration (Suppl. Fig. 14).

The neuro-inflammatory data has been shown in Suppl fig 15.

3. Also, are the astrocytes functionally active?

• Our data show strong responses of the astrocytes in the AstTau model. These changes include expression of reactive astrocytic marker GFAP (Fig. 1b). The astrocytes also exhibit strong astrogliosis and extension of processes in AstTau, which further indicates functional activity (Suppl. Fig. 6).

Extension of processes not possible to observe and this claim should be removed. Increase in GFAP is clear and should be noted in the main text.

4. Supplementary video 1 shows a single neuron with minimal branching and it is difficult to see the arborization and interactions claimed by the authors. Neuron-astrocyte co-cultures have been performed previously in 2D based on the video provided, it cannot be claimed that the 3D cultures results in a more mature phenotype.

• Thank you for the feedback on the videos, we have improved this analysis with high resolution imaging of 10 μm sections. This high-resolution imaging shows the intimate proximity of the neurons and astrocytes, which complements the extensive arborization and interactions shown in Suppl. Fig. 2b, that develop in the AstTau model as compared to 2D culture (Suppl. Fig. 2).

• We also show a comparison of arborization in 2D vs. 3D cultures (Suppl. Fig. 2a) that demonstrates the strong increase in arborization upon growth in 3D.

The 2D data shown is at 1 week while the immunostaining data shown is at 21 DIV. These are not comparable. The videos provided do not support the claims and should be removed in my opinion. The filled neurons in Suppl Fig 4 support the claims regarding arborization better. But we see this level of complexity in 2D cultures as well. Based on the data provided, I am not convinced that the 3D asteroids generate a more mature phenotype.

6. In Fig. 1f, the inhibitory neuronal population seems stable in the 3D cultures but the excitatory neuronal populations keep increasing from as low as ~10% at 7 DIV3D to ~40% by 21 DIV3D. This is unexpected because NGN2 induces mainly excitatory neurons from NPCs. So I would have expected a larger percentage of excitatory neurons compared to inhibitory neurons in their starting cultures. This also means that a significant proportion of cells at the phenotypic time point are immature as they are newly born. It is not clear whether the phenotypes observed are due to the immature or mature population. The authors measure maturity but it is unclear how this was measured.

• The reviewer has astutely noted classic points stemming from the nature of iPSC culture modeling. One of the strengths of the AstTau system is that it does indeed generate multiple neuronal cell types (including inhibitory neurons). While the precise mechanism responsible for this technological advance remains to be determined, it seems possible that the increased phenotypic diversity results from the use of partial NGN2 induction and the subsequent integration with astrocytes in 3D culture. We also note that NGN2 overexpression has recently been demonstrated to produce a broader range of neuronal phenotypes than originally thought¹, which we are able to more finely dissect with scRNA-seq. Our data further show that the immature precursor population that is initially present in AstTau undergo further maturation, with this immature population dramatically shrinking over the 21 DIV3D time course, resulting in more mature neurons that are able to recapitulate neurodegeneration (Fig. 1, Suppl. Fig 5).

Maybe I missed it and apologies if I did, but it is not clear how maturity was measured.

10. Phenotypic analysis needs to be expanded. An early hallmark of AD is the loss of synapses. Can the authors recapitulate this phenotype in their 3D culture?

• This is an important additional to the phenotypic analysis, thank you for the suggestion. We have now performed VGLUT1 immunolabeling and quantified glutamatergic synaptic loss. The resulting data provide additional information supporting the utility of our system in recapitulation neurodegeneration (Suppl. Fig. 10 d-e).

VGLUT1 alone is not a marker of synapses. But loss of VGLUT1 does indicate loss of excitatory neurons. Please remove the claim of synaptic loss and indicate that the VGLUT1 data represents a loss of excitatory neurons.

12. Are astrocytes even required to propagate the phenotype? The S100b staining would suggest that gliosis occurs earlier than the Fluro Jade staining but is that a technical artefact of detection limits or is astrocytic pathology required or does it accelerate neuronal dysfunction? Do astrocytes also show oTau pathology? And at what stage?

• Astrocytes are necessary to the production and continuity of the 3D AstTau model (Suppl. Fig. 2). Indeed, it is not possible to study the system without astrocytes because the neurons do not maintain a viable 3D structure. We do note that astrocyte pathology precedes neurodegeneration by morphological changes and S100 β staining, adding to the growing body of evidence that astrocytes play a critical role in the propagation of neurodegeneration phenotypes. The AstTau model is a powerful platform that enables dissection of astrocytic contributions, which we demonstrate in this manuscript.

The observation that astrogliosis is observed early in the degenerative process is an

important point and should be highlighted in the discussion section.

18. Do they observe enrichment of familial AD associated genes in the upregulated or downregulated gene sets?

• **Really interesting question! Since the AstTau model is focused on the contribution of oTau to the development of pathology in non-familial AD iPSCs we had not performed that analysis. At your suggestion we have mapped expression changes of familial AD associated GWAS genes between conditions in AstTau at 21 DIV3D (Fig. 7d).**

This figure showing this analysis seems to be missing.

19. Since they have the single cell profiles and a measure of maturity, do they see a correlation of the observed phenotypes with the maturity of the neurons? Can they order the neurons along a pseudotime of degeneration either by using all the differentially expressed genes or genes involved in apoptosis?

• **Thank you for this suggestion, this is a fantastic way to broaden our analysis of neurodegeneration in our dataset we had not tried. We have added this analysis and discussion in the manuscript (Fig 8).**

Interesting analysis though the panel labels and figure legends do not match.

22. Also, can they highlight the figure that shows a progressive loss of glutamatergic neurons?

• **We have added a figure to highlight this loss and associated pathways! (Fig 8, Suppl. Fig. 10d-f).**

Suppl 10f does not show a progressive loss. I agree the data does indicate a loss of VGLUT1 neurons at DIV 21. The authors need to remove the claim of a "progressive" loss.

Reviewer #2 (Remarks to the Author):

The revised manuscript is significantly strengthened and addresses several points raised by the reviewers. Some minor weaknesses remain and should be addressed before publication:

- Some of the staining shown still raises concerns about either antibody specificity or the biology of the system: E.g. Figure 1 NEUN is not nuclear; some of the MAP2 stained neurons overlap with SOX2

- While they authors say they addressed the cell type specificity problem, it seems that they didn't actually segment the MC1+, Tuj1+ cells, but just normalized according to total Tuj1 intensity

RESPONSES TO REVIEWERS' COMMENTS

August 19, 2022

To the reviewers:

We thank the reviewers for their careful consideration and critique of our revised manuscript. We have addressed all your remaining concerns and present a description of our responses below.

Reviewer #1 (Remarks to the Author):

The authors have addressed most of my concerns either through experiments or clarifications. There are a few minor changes regarding some claims I would like to see before acceptance.

1. The single cell analysis also did not reveal anything significantly more than what was already known. The role of chaperone proteins, especially the heat shock proteins is well known in propagating Tau pathology. HSP90 inhibition has been found to reduce Tau pathology almost ten years ago (PMID: 25069659).

- You are absolutely correct, heat shock chaperones proteins have been well established to play a role in tau mediated neurodegeneration, even recently highlighted by single cell RNA sequencing of post-mortem AD brain tissue, so we were excited to see the heat shock response in our system as it demonstrated that our model is accurately recapitulating the processes associated with neurodegeneration as our goal is to produce a model that recapitulates AD and is responsive to neuroprotective treatments. We also demonstrated neuroinflammatory states that differed between excitatory and inhibitory neurons and in subsequent analysis have identified glutamatergic neuron and astrocyte subpopulation signatures that are associated with neurodegeneration (Suppl. Fig. 14).

The neuro-inflammatory data has been shown in Suppl fig 15.

-The reviewer is correct, the neuro-inflammatory data is shown in Suppl. Fig. 15 and should have been referenced in this response. Suppl. Fig. 14 as written is referencing the additional analysis identifying pathology associated astrocyte subpopulation signatures.

3. Also, are the astrocytes functionally active?

- Our data show strong responses of the astrocytes in the AstTau model. These changes include expression of reactive astrocytic marker GFAP (Fig. 1b). The astrocytes also exhibit strong astrogliosis and extension of processes in AstTau, which further indicates functional activity (Suppl. Fig. 6).

Extension of processes not possible to observe and this claim should be removed. Increase in GFAP is clear and should be noted in the main text.

-On the reviewer's recommendation we have removed this claim on page 14 and have instead emphasized the notable increase in GFAP we observe in AstTau.

4. Supplementary video 1 shows a single neuron with minimal branching and it is difficult to see the arborization and interactions claimed by the authors. Neuron-astrocyte co-cultures have been performed previously in 2D based on the video provided, it cannot be claimed that the 3D cultures results in a more mature phenotype.

- Thank you for the feedback on the videos, we have improved this analysis with high resolution imaging of 10 μm sections. This high-resolution imaging shows the intimate proximity of the neurons and astrocytes, which complements the extensive arborization and interactions shown in Suppl. Fig. 2b, that develop in the AstTau model as compared to 2D culture (Suppl. Fig. 2).
- We also show a comparison of arborization in 2D vs. 3D cultures (Suppl. Fig. 2a) that demonstrates the strong increase in arborization upon growth in 3D.

The 2D data shown is at 1 week while the immunostaining data shown is at 21 DIV. These are not comparable. The videos provided do not support the claims and should be removed in my opinion. The filled neurons in Suppl Fig 4 support the claims regarding arborization better. But we see this level of complexity in 2D cultures as well. Based on the data provided, I am not convinced that the 3D asteroids generate a more mature phenotype.

-We agree that the videos do not support the claims as well as other supplemental figures (e.g., Suppl. Fig. 1, 2, and 4). Therefore, the videos have been removed from the manuscript. Our comment that 3D asteroids generate a more mature phenotype than 2D cultures is an observation that has also been observed and published by other groups previously; to help clarify this issue, we have now included references 26 and 27, which provide literature support for our observation that 3D cultures provide more complexity than 2D cultures.

26. Venkataraman, L., Fair, S.R., McElroy, C.A., Hester, M.E. & Fu, H. Modeling neurodegenerative diseases with cerebral organoids and other three-dimensional culture systems: focus on Alzheimer's disease. Stem Cell Reviews and Reports (2020).

27. Choi, S.H., Kim, Y.H., Quinti, L., Tanzi, R.E. & Kim, D.Y. 3D culture models of Alzheimer's disease: a road map to a "cure-in-a-dish". Molecular neurodegeneration 11, 75-75 (2016).

6. In Fig. 1f, the inhibitory neuronal population seems stable in the 3D cultures but the excitatory neuronal populations keep increasing from as low as ~10% at 7 DIV3D to ~40% by 21 DIV3D. This is unexpected because NGN2 induces mainly excitatory neurons from NPCs. So I would have expected a larger percentage of excitatory neurons compared to inhibitory neurons in their starting cultures. This also means that a significant proportion of cells at the phenotypic time point are immature as they are newly born. It is not clear whether the phenotypes observed are due to the immature or mature population. The authors measure maturity but it is unclear how

this was measured.

- The reviewer has astutely noted classic points stemming from the nature of iPSC culture modeling. One of the strengths of the AstTau system is that it does indeed generate multiple neuronal cell types (including inhibitory neurons). While the precise mechanism responsible for this technological advance remains to be determined, it seems possible that the increased phenotypic diversity results from the use of partial NGN2 induction and the subsequent integration with astrocytes in 3D culture. We also note that NGN2 overexpression has recently been demonstrated to produce a broader range of neuronal phenotypes than originally thought¹, which we are able to more finely dissect with scRNA-seq. Our data further show that the immature precursor population that is initially present in AstTau undergo further maturation, with this immature population dramatically shrinking over the 21 DIV3D time course, resulting in more mature neurons that are able to recapitulate neurodegeneration (Fig. 1, Suppl. Fig 5).

Maybe I missed it and apologies if I did, but it is not clear how maturity was measured.

-Apologies for not making the references clearer in our response. Maturity was measured by quantification of immunolabeling and scRNA-seq gene expression of markers of developing and mature cell types (Fig. 1), as well as through integration and comparison with previously published scRNA-seq cerebral organoid datasets (Fig 1, Suppl. Fig. 5). This is discussed on the bottom of page 5:

“To compare the maturity of neurons and astrocytes in our asteroid culture system to traditional cerebral organoid models we integrated our asteroid scRNA-seq data with an existing published scRNA-seq data set of forebrain cerebral organoids generated using three previously published protocols with different levels of directed differentiation³⁸. Clustering, cell type identification, and cluster composition analysis was performed on the new integrated dataset and revealed that the asteroid model (red) was overrepresented in more mature cell type clusters compared to the cerebral organoid model (blue) (Fig 1j, Suppl. Fig. 7).”

10. Phenotypic analysis needs to be expanded. An early hallmark of AD is the loss of synapses. Can the authors recapitulate this phenotype in their 3D culture?

- This is an important additional to the phenotypic analysis, thank you for the suggestion. We have now performed VGLUT1 immunolabeling and quantified glutamatergic synaptic loss. The resulting data provide additional information supporting the utility of our system in recapitulation neurodegeneration (Suppl. Fig. 10 d-e).

VGLUT1 alone is not a marker of synapses. But loss of VGLUT1 does indicate loss of excitatory neurons. Please remove the claim of synaptic loss and indicate that the VGLUT1 data represents a loss of excitatory neurons.

-Thank you for pointing this out, we agree. We have changed the terms to focus on the loss of excitatory neurons evidenced by VGLUT1 loss in the text and supplementary figures.

Additionally, we have added a new panel to highlight the loss of synaptic gene markers by scRNA-seq (Suppl. Fig. 10i) (bottom of Page 7).

“Notably, neurodegeneration in AstTau is associated with a loss of excitatory neurons as evidenced by scRNA-seq cell type composition analysis and 25% loss of VGLUT1 immunolabeling, as is observed in clinical tauopathy progression (Suppl. Fig. 10d-f). There is also a striking reduction in synaptic gene markers by scRNA-seq in AstTau (Suppl. Fig. 10i).”

12. Are astrocytes even required to propagate the phenotype? The S100b staining would suggest that gliosis occurs earlier than the Fluro Jade staining but is that a technical artefact of detection limits or is astrocytic pathology required or does it accelerate neuronal dysfunction? Do astrocytes also show oTau pathology? And at what stage?

- Astrocytes are necessary to the production and continuity of the 3D AstTau model (Suppl. Fig. 2). Indeed, it is not possible to study the system without astrocytes because the neurons do not maintain a viable 3D structure. We do note that astrocyte pathology precedes neurodegeneration by morphological changes and S100 β staining, adding to the growing body of evidence that astrocytes play a critical role in the propagation of neurodegeneration phenotypes. The AstTau model is a powerful platform that enables dissection of astrocytic contributions, which we demonstrate in this manuscript.

The observation that astrogliosis is observed early in the degenerative process is an important point and should be highlighted in the discussion section.

-Thank you for this suggestion, we agree the early astrogliosis is very notable! We have added this discussion on page 14/15.

“AstTau exhibits remarkable astrocytic stress responses resulting in expansion of the astrocytic compartment, selective transcriptional responses, and changes consistent with reactive astrogliosis, including increased GFAP expression. This coordinated stress response is observed as soon as 7 DIV3D, indicating rapid and coordinated crosstalk between neurons exposed to tau oligomers and the otherwise healthy astrocytes.”

18. Do they observe enrichment of familial AD associated genes in the upregulated or downregulated gene sets?

- Really interesting question! Since the AstTau model is focused on the contribution of oTau to the development of pathology in non-familial AD iPSCs we had not performed that analysis. At your suggestion we have mapped expression changes of familial AD associated GWAS genes between conditions in AstTau at 21 DIV3D (Fig. 7d).

This figure showing this analysis seems to be missing.

-Apologies, the reference in the reviewer response was incorrect. The correct reference is for the panel is Fig. 5d

19. Since they have the single cell profiles and a measure of maturity, do they see a correlation of the observed phenotypes with the maturity of the neurons? Can they order the neurons along a pseudotime of degeneration either by using all the differentially expressed genes or genes involved in apoptosis?

- Thank you for this suggestion, this is a fantastic way to broaden our analysis of neurodegeneration in our dataset we had not tried. We have added this analysis and discussion in the manuscript (Fig 8).

Interesting analysis though the panel labels and figure legends do not match.

-Thank you for noting this, we have reviewed the figure and figure legend to ensure they match.

22. Also, can they highlight the figure that shows a progressive loss of glutamatergic neurons?

- We have added a figure to highlight this loss and associated pathways! (Fig 8, Suppl. Fig. 10d-f).

Suppl 10f does not show a progressive loss. I agree the data does indicate a loss of VGLUT1 neurons at DIV 21. The authors need to remove the claim of a “progressive” loss.

-Thank you for the input, we have removed this claim.

Reviewer #2 (Remarks to the Author):

The revised manuscript is significantly strengthened and addresses several points raised by the reviewers. Some minor weaknesses remain and should be addressed before publication:

- Some of the staining shown still raises concerns about either antibody specificity or the biology of the system: E.g. Figure 1 NEUN is not nuclear; some of the MAP2 stained neurons overlap with SOX2

- Thank you for raising this question. We agree that it is difficult to completely separate the cell population by immunofluorescence labeling. The reason appears to arise from both the method itself and the biology of the system. First, the Asteroids are quite small; they are only 100µm thick. We did not section the Asteroids because of the small size, which means that the immunofluorescence is done on the entire Asteroid. Analysis of the entire Asteroid results in some overlap of fluorescence signal from cells in different planes (even though we used confocal imaging). Secondly, some neuronal cells might be on the transition between progenitor cells to mature neurons and it is possible that they present both SOX2 and MAP2.

The reviewer is correct that some NeuN is present in neuronal cell bodies. This is actually something that is commonly observed with iPS derived neurons and is not unexpected because NeuN is an RNA binding protein which therefore has the capability of being both nuclear and cytoplasmic. The biology of iPS derived neurons appears to result in the presence of some NeuN in soma of neurons.

- While they authors say they addressed the cell type specificity problem, it seems that they didn't actually segment the MC1+, Tuj1+ cells, but just normalized according to total Tuj1 intensity

-We apologize for not completely addressing your question in last revision. We have included new panels to present the raw fluorescence intensity of MC1 and Tuj1 separately to segment MC1+, Tuj1+ cells in the current revision (Suppl. Fig. 9b-c). Additionally, we have quantified the colocalization ratio of MC1 to Tuj1 and S100b respectively to address cell type specific tau pathology (Suppl. Fig. 9d).

REVIEWERS' COMMENTS

Reviewer #1 (Remarks to the Author):

The authors have addressed all my concerns. I recommend this manuscript for publication.

Reviewer #2 (Remarks to the Author):

The authors have addressed my remaining concerns, and the manuscript is suitable for publication.